# THE AUGMENTED IMAGE PRIOR: DISTILLING 1000 CLASSES BY EXTRAPOLATING FROM A SINGLE IMAGE

**Yuki M. Asano**[*]
Video & Image Sense Lab
University of Amsterdam

**Aaqib Saeed**[*]
Department of Industrial Design
Eindhoven University of Technology

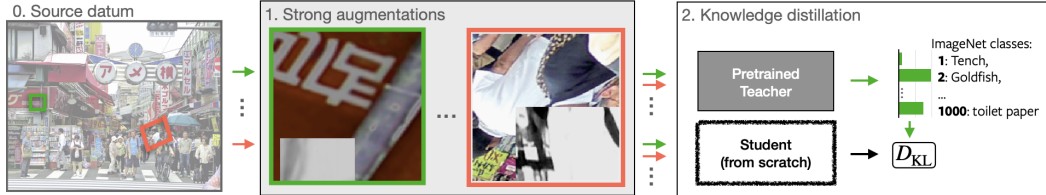

Figure 1: **Extrapolating from one image.** Strongly augmented patches from a single image are used to train a student (S) to distinguish semantic classes, such as those in ImageNet. The student neural network is initialized randomly and learns from a pretrained teacher (T) via KL-divergence. Although almost none of target categories are present in the image, we find student performances of $>69\%$ for classifying ImageNet's 1000 classes. In this paper, we develop this single datum learning framework and investigate it across datasets and domains.

## ABSTRACT

What can neural networks learn about the visual world when provided with only a single image as input? While any image obviously cannot contain the multitudes of all existing objects, scenes and lighting conditions – within the space of all $256^{3 \cdot 224 \cdot 224}$ possible 224-sized square images, it might still provide a strong prior for natural images. To analyze this "augmented image prior" hypothesis, we develop a simple framework for training neural networks from scratch using a single image and augmentations using knowledge distillation from a supervised pretrained teacher. With this, we find the answer to the above question to be: 'surprisingly, a lot'. In quantitative terms, we find accuracies of $94\%/74\%$ on CIFAR-10/100, $69\%$ on ImageNet, and by extending this method to video and audio, $51\%$ on Kinetics-400 and $84\%$ on SpeechCommands. In extensive analyses spanning 13 datasets, we disentangle the effect of augmentations, choice of data and network architectures and also provide qualitative evaluations that include lucid "panda neurons" in networks that have never even seen one. Code: https://single-image-distill.github.io/

## 1 INTRODUCTION

Deep learning has both relied and improved significantly with the increase in dataset sizes. In turn, there are many works that show the benefits of dataset scale in terms of data points and modalities used. Within computer vision, these models trained on ever larger datasets, such as Instagram-1B (Mahajan et al., 2018) or JFT-3B (Dosovitskiy et al., 2021), have been shown to successfully distinguish between semantic categories at high accuracies. In stark contrast to this, there is little research on understanding neural networks trained on *very small* datasets. Why would this be of any interest? While smaller dataset sizes allow for better understanding and control of what the model is being trained with, we are most interested in its ability to provide insights into fundamental aspects of learning: For example, it is an open question as to what exactly is required for arriving at semantic

---

[*]Equal contribution.

visual representations from random weights, and also of how well neural networks can extrapolate from their training distribution.

While for visual models it has been established that few or no real images are required for arriving at basic features, like edges and color-contrasts (Asano et al., 2020; Kataoka et al., 2020; Bruna & Mallat, 2013; Olshausen & Field, 1996), we go far beyond these and instead ask what the minimal data requirements are for neural networks to learn *semantic categories*, such as those of ImageNet. This approach is also motivated by studies that investigate the early visual development in infants, which have shown how little visual diversity babies are exposed to in the first few months whilst developing generalizeable visual systems (Orhan et al., 2020; Bambach et al., 2018). In this paper, we study this question in its purest form, by analyzing whether neural networks can learn to extrapolate from a single datum.

However, addressing this question naïvely runs into the difficulties of i) current deep learning methods, such as SGD or BatchNorm being tailored to large datasets and not working with a single datum and ii) extrapolating to semantic categories requiring information about the space of natural images beyond the single datum. In this paper, we address these issues by developing a simple framework that recombines *augmentations* and *knowledge distillation*.

First, augmentations can be used to generate a large number of variations from a single image. This can effectively address issue i) and allow for evaluating the research question on standard architectures and datasets. This use of data augmentations to generate variety is drastically different to their usual use-case in which transformations are generated to implicitly encode desirable invariances during training.

Second, to tackle the difficulty of providing information about semantic categories in the single datum setting, we opt to use the outputs of a supervisedly trained model in a knowledge distillation (KD) fashion. While KD (Hinton et al., 2015) is originally proposed for improving small models' performance by leveraging what larger models have learned, we re-purpose this as a simple way to provide a supervisory signal about semantic classes into the training process.

We combine the above two ideas and provide both student and teacher only with augmented versions of a single datum, and train the student to match the teacher's imagined class-predictions of classes – almost all of which are not contained in the single datum, see Fig. 1. While practical applications do result from our method – for example we provide results on single image dataset based model compression in the Appendix – our goal in this paper is analyzing *the fundamental question* of how well neural networks trained from a single datum can extrapolate to semantic classes, like those of CIFAR, SpeechCommands, ImageNet or even Kinetics.

What we find is that despite the fact that the resulting model has only seen a single datum plus augmentations, surprisingly high quantitative performances are achieved: *e.g.* 74% on CIFAR-100, 84% on SpeechCommands and 69% top-1, single-crop accuracy on ImageNet-12. We further make the novel observations that our method benefits from high-capacity student and low-capacity teacher models, and that the source datum's characteristics matter – random noise or less dense images yield much lower performances than dense pictures like the one shown in Figure 1.

In summary, in this paper we make these four main contributions:

1. A minimal framework for training neural networks with a single datum using distillation.
2. Extensive ablations of the proposed method, such as the dependency on the source image, augmentations and network architectures.
3. Large scale empirical evidence of neural networks' ability to extrapolate on $> 12$ vision and audio datasets.
4. Qualitative insights on what and how neural networks trained with a single image learn.

## 2 RELATED WORK

The work presented builds on top of insights from the topics of knowledge distillation and single- and no-image training of visual representations and yields insights into neural networks' ability to extrapolate.

**Distillation.** In the original formulation of knowledge distillation (KD), the goal is to train a typically lower capacity ("student") model from a pretrained ("teacher") model in order to surpass the performance of solely training with a label-supervised objective. KD has also been explored extensively to train more performant and, or compressed student models from the soft-target predictions of teacher models (Ba & Caruana, 2014; Hinton et al., 2015; Gou et al., 2021; Furlanello et al., 2018; Czarnecki et al., 2017). Similarly, other approaches have also been developed to improve transfer from teacher to student, including sharing intermediate layers' features (Romero et al., 2014), spatial attention transfer (Zagoruyko & Komodakis, 2016b), similarity preservation between activations of the networks (Tung & Mori, 2019), contrastive distillation (Tian et al., 2020a) or few-shot distillation (Li et al., 2020). KD has also been shown to be an effective approach for learning from noisy labels (Li et al., 2017). More recently, Beyer et al. (2022) conducted a comprehensive empirical investigation to identify important design choices for successful distillation from large-scale vision models. In particular, they show that long training schedules, paired with consistent augmentations (including *MixUp*) for both student and teacher, result in better performances.

**Without the training data.** Distilling knowledge without access to the original training dataset was originally proposed in 2017 (Lopes et al., 2017) and first experiments of leaving out single MNIST classes from the training data were shown in (Hinton et al., 2015). Yet, this paradigm is gaining importance as many recent advancements have been made possible due to extremely large proprietary datasets that are kept private. This has lead to either the sole release of trained models (Radford et al., 2021; Ghadiyaram et al., 2019; Mahajan et al., 2018) or even more restricted access to only the model outputs via APIs (Brown et al., 2020). While the original work (Lopes et al., 2017), still required activation statistics from the training dataset of the network, more recent works do not require this "meta-data". These approaches are typically generation based (Chen et al., 2019; Ye et al., 2020; Micaelli & Storkey, 2019; Yin et al., 2020) and *e.g.* yield datasets of synthetic images that maximally activate neurons in the final layer of teacher. Related to this, there are works which conduct "dataset" distillation, where the objective is to distill large-scale datasets into much smaller ones, such that models trained on it reach similar levels of performance as on the original data. These methods generate synthetic images (Wang et al., 2018; Radosavovic et al., 2018; Liu et al., 2019; Zhao & Bilen, 2021; Cai et al., 2020), labels (Bohdal et al., 2020), or both (Nguyen et al., 2021), but due to their meta-learning nature have only been successfully applied to small-scale datasets. In contrast to GAN- and inversion-based methods, as well as to dataset distillation, our approach does not require the knowledge of the weights and architecture of the teacher model, and instead works with black-box "API"-style teacher models and much smaller "datasets" of just a single datum plus augmentations. This paper follows a radically different goal: We are interested in *analysing* the potential of extrapolating from a single image to the manifold of natural images, for which we choose KD as a well-suited and simple tool.

**Prior knowledge in deep learning.** Finally, this work is related to several works which have analysed the infusion of prior knowledge to the training process. For example the prior knowledge contained in neural network architectures (Jarrett et al., 2009; Zhou et al., 2019; Sreenivasan et al., 2021; Kim et al., 2021; Baek et al., 2021; Ulyanov et al., 2018) or image augmentations (Xiao et al., 2020). In this work, we instead view a single image as a "prior" for all other natural images, which to the best of our knowledge has not been explored.

## 3 METHOD

We believe that simplicity is the key to demonstrating and analysing the question of how far a single image can take us. We are inspired by recent work of Asano et al. (2020), which "patchifies" a single-image using augmentations, and the knowledge distillation method presented by Beyer et al. (2022) and our technical contribution lies in unifying them to a *single-image distillation* framework.

**i. Dataset generation.** In (Asano et al., 2020), a single "source" image is augmented many times to generate a static dataset of fixed size. This is done by applying the following augmentations in sequence: cropping, rotation and shearing, and color jittering. For this, we follow their official implementation Asano et al. (2020), and do not change any hyperparameters. We do however analyze the choice of source images more thoroughly by additionally including a random noise, and a Hubble-telescope image to our analysis. In addition, we also conduct experiments on audio classification. For generating a dataset of augmented audio-clips, we apply the set of audio-augmentations from (Bitton & Papakipos, 2021), which consist of operations, such as random volume increasing,

Table 1: **Distilling dataset.** 1 image + augmentations ≈ almost 50K in-domain CIFAR-10/100 images. Here, both teacher and student are WideR40-4 networks.

| Distillation dataset | | | | Accuracy | |
|---|---|---|---|---|---|
| Name | # Images | # Pixels | Size (MB) | C10 | C100 |
| CIFAR-10 | 50K | 51M | 145 | 95.26 | 76.29 |
| CIFAR-100 | 50K | 51M | 145 | 94.51 | 78.06 |
| CIFAR-10 | 10K | 10M | 29 | 94.58 | 72.95 |
| Ours | 1 | 2.8M | 0.27 | 94.14 | 73.80 |

Table 2: **Comparison to other datasets.** Teacher model is WideR40-2 and student is WideR40-1 as in (Micaelli & Storkey, 2019).

| Data | C10 |
|---|---|
| CIFAR-10 | 92.61 |
| Fractals | 33.26 |
| StyleGAN | 83.42 |
| ZeroSKD | 86.60 |
| Ours | 89.27 |

background noise addition and pitch shifting to yield log-Mel spectrograms from raw waveforms (see Appendix for complete details).

**ii. Knowledge distillation.** The knowledge distillation objective is proposed in (Hinton et al., 2015) to transfer the knowledge of a pretrained teacher to a lower capacity student model. In this case, the optimization objective for the student network is a weighted combination of dual losses: a standard supervised cross-entropy loss and a "distribution-matching" objective that aims to mimic the teacher's output. However, in our case there are no class-labels for the patches generated from a single image, so we solely use the second objective formulated as a Kullback–Leibler (KL) divergence between the student output $p^s$ and the teacher's output $p^t$:

$$\mathcal{L}_{\mathrm{KL}} = \sum_{c \, \in \, \mathcal{C}} -p_c^t \log p_c^s + p_c^t \log p_c^t \tag{1}$$

where $c$ are the teachers' classes and the outputs of both student and teacher are temperature $\tau$ flattened probabilities, $p = \texttt{softmax}(l/\tau)$, that are generated from logits $l$.

For training, we follow (Beyer et al., 2022) in employing a function matching strategy, where the teacher and student models are fed consistently augmented instances, that include heavy augmentations, such as *MixUp* (Zhang et al., 2018) or *CutMix* (Yun et al., 2019). However, in contrast to (Beyer et al., 2022), we neither have access to TPUs nor can train 10K epochs on ImageNet-sized datasets. While both of these would likely improve the quantitative results, we believe that this handicap is actually blessing in disguise: This means that the results we show in this paper are not specific to heavy-compute, or extremely large batch-sizes, but instead are fundamental.

## 4 EXPERIMENTS

### 4.1 IMPLEMENTATION DETAILS

**Source data.** For the source data, we utilize the single images of (Asano et al., 2020), except for one image, which we replace by a similar one as we could not retrieve its licence. These images are of sizes up to 2,560x1,920 (the "City" image of Fig. 1). For the audio experiments, we use two short audio-clips from Youtube, a 5mins BBC newsclip, as well as a 5mins clip showing 11 Germanic languages. Images and audio visualizations, sources and licences are provided in Appendix A.

**Tasks.** For simplicity, we focus on classification tasks here and provide results on a sample application of single-image based model compression in Appendix B.10. For our ablations and small-scale experiments we focus on CIFAR-10/100 (Krizhevsky et al., 2009). For larger-scale experiments using $224 \times 224$ sized images, we evaluate our method on datasets with varying number of classes (see Table 7) and additionally conduct experiments in the audio domain (see Table 5) and video (see Table 6). Further implementation details are provided in Appendix C.

### 4.2 ABLATIONS

**1-image vs full training set.** We first examine the capability to extrapolate from one image to small-scale datasets, such as CIFAR-10 and CIFAR-100. In Table 1, we compare various datasets for distilling a teacher model into a student. We find that while distillation using the source dataset always works best (95.26% and 78.06%) on CIFAR-10/100, using a single image can yield models

Table 3: **Ablations of single image distillation.** We analyze key components in our experimental setup. We report top-1 accuracies for CIFAR-10/100. For the teacher we use WideR40-4 and for the student a WideR16-4 and train for 1K epochs.

| Distillation dataset | | Accuracy | |
|---|---|---|---|
| Image | # Pixels | C10 | C100 |
| "Noise" | 4.9M | 69.30 | 19.50 |
| "Universe" | 4.8M | 88.18 | 39.68 |
| "Bridge" | 1.1M | 92.24 | 57.87 |
| "City" | 4.9M | 93.13 | 64.85 |
| "Animals" | 2.8M | 93.28 | 66.12 |

(a) **Distilling image.** Content of source image matters.

| | Accuracy | |
|---|---|---|
| Signal | C10 | C100 |
| Full | 93.32 | 68.69 |
| Top-5 | 92.98 | 64.72 |
| Argmax | 91.89 | 60.75 |

(b) **Teacher signal.** Even with only top-5, or hard distillation, performance only slightly degrades.

| Augmentations | | | | Accuracy | |
|---|---|---|---|---|---|
| Hflip. | RCrop. | MixUp | CutMix | C10 | C100 |
| ✓ | ✓ | | | 89.34 | 55.05 |
| ✓ | | ✓ | | 91.03 | 58.24 |
| ✓ | | | ✓ | 92.86 | 64.26 |
| ✓ | ✓ | ✓ | | 92.41 | 63.50 |
| ✓ | ✓ | | ✓ | 93.32 | 68.69 |

(c) **Varying Augmentation.** More helps, CutMix is important.

Figure 2: **Varying source images** for CIFAR-100 distillation. Setting as in 3a.

which almost reach this upper bar (94.1% and 73.8%). Moreover, we find that one image distillation even outperforms using 10K images of CIFAR-10 when teaching CIFAR-100 classes even though these two datasets are remarkably similar. To better understand why using a single image works, we next disentangle the various components used in the training procedure: (a) the source image, (b) the generated image dataset size and (c) the augmentations used during distillation, corresponding to Tables 3a and 3c.

**Choice of single image.** From Table 3a, we find that the choice of source image content is crucial: Random noise or sparser images perform significantly worse compared to the denser "City" and "Animals" pictures. We compare 23 further images in Fig. 2 and find that distillation quality roughly correlates with the density in images and JPEG sizes. This is in contrast to self-supervised pretraining of (Asano et al., 2020), suggesting that the underlying mechanisms are different.

**Varying loss functions.** In Table 3b, we show that the student can learn even with much degraded learning signals. For example even if it receives only the top-5 predictions or solely the argmax prediction (*i.e.* a hard label) of the teacher without any confidence value, the student is still able to extrapolate at a significant level ($> 91\%/60\%$ for CIFAR-10/100). Even at ImageNet scale, the performance remains at a high 43.8% top-1 accuracy (see Table 7 row *(m)*). This also suggests that copying models solely from API outputs is possible, similar to (Orekondy et al., 2019). We also evaluated L1 and L2 distillation functions (see Table 9 for details) and found that these perform slightly worse. This echoes the finding of (Tian et al., 2020a) that standard KD loss of Eq. (1) is actually a strong baseline, hence we use this in the rest of our experimental analysis.

**Augmentations.** In Table 3c, we ablate the augmentations we use during knowledge distillation. Besides observing the general trend of "more augmentations are better", we find that *CutMix* performs better than MixUp on our single-image distillation task. This might be because in our case the model is tasked with learning how to extrapolate towards real datapoints, while MixUp is derived and therefore might be more useful for interpolating *between* (real) data samples (Zhang et al., 2018; Beyer et al., 2022).

**Comparison to synthetic datasets.** In Table 2, we find that there is something unique about using a single image, as our method outperforms several synthetic datasets, such as FractalDB Kataoka et al. (2020), randomly initialized StyleGAN Baradad et al. (2021), as well as the GAN-based ap-

Table 4: **Distilling various architectures** on CIFAR-10/100. We compare student accuracy when distilling with full training set vs our 1-image dataset

|  | Teacher | Acc. | Student | Acc. | Full | Ours | $\Delta < 5\%$ |
|---|---|---|---|---|---|---|---|
| | | | | | | **Distillation** | |
| CIFAR10 | VGG-19 | 93.28 | VGG-16 | 92.42 | 92.84 | 92.14 | ✓ |
| | ResNet-56 | 93.77 | ResNet-20 | 92.52 | 92.29 | 90.70 | ✓ |
| | WideR40-4 | 95.42 | WideR16-4 | 95.20 | 95.00 | 93.32 | ✓ |
| | WideR40-4 | 95.42 | WideR40-4 | 95.42 | 94.36 | 94.14 | ✓ |
| | WideR16-4 | 95.20 | WideR40-4 | 95.42 | 94.30 | 94.02 | ✓ |
| CIFAR100 | VGG-19 | 70.79 | VGG-16 | 73.26 | 71.19 | 58.66 | ✗ |
| | ResNet-56 | 70.99 | ResNet-20 | 65.74 | 67.04 | 52.43 | ✗ |
| | WideR40-4 | 78.14 | WideR16-4 | 75.56 | 76.26 | 68.69 | ✗ |
| | WideR40-4 | 78.14 | WideR40-4 | 78.14 | 75.54 | 73.80 | ✓ |
| | WideR16-4 | 78.14 | WideR40-4 | 75.56 | 76.29 | 74.08 | ✓ |

Table 5: **Distilling audio representations.** 1 audio clip + augmentations provide stronger supervisory signal for the student to distill teacher's knowledge.

| Dataset | Categories (#Classes) | Teacher | Full | Ours$_A$ | Ours$_B$ |
|---|---|---|---|---|---|
| | | | | **Distillation** | |
| MUSAN (Snyder et al., 2015) | Generic sounds (3) | 98.76 | 96.78 | 89.85 | 96.28 |
| Voxforge (MacLean, 2018) | Languages (6) | 91.13 | 89.04 | 72.85 | 78.47 |
| Speech Commands (Warden, 2018) | Keywords (12) | 95.15 | 94.86 | 82.90 | 84.19 |
| LibriSpeech (Panayotov et al., 2015) | Speakers (100) | 99.89 | 99.54 | 78.67 | 84.12 |

proach of (Micaelli & Storkey, 2019). This is despite the fact that these synthetic datasets contain $\sim 50K$ images. We provide insights on the characteristics of this 1-image dataset in Section 4.4. Next, using the insights gained in this section, we scale the experiments towards other network architectures, dataset domains and dataset sizes.

## 4.3 GENERALISATION

**Architectures** In Table 4, we experiment with different common architectures on CIFAR-10 and CIFAR-100. On CIFAR-10, we find that almost all architectures perform similarly well, except for the case of ResNet-56 to ResNet-20 which could be attributed to the lower capacity of the student model. We find that the distillation performance of WideR40-4 to WideR40-4 on CIFAR-10 is very close to distilling from original source data; achieving accuracy of $94.14\%$ and only around one-percentage point lower than supervised training. This analysis shows that our method generalizes to other architectures hints that larger capacity student models might be important for high performances.

**Extension to audio.** So far, our proposed approach has shown success in distilling useful knowledge for images. To further test the generalizability of our approach on other modalities, we conduct experiments on several audio recognition tasks of varying difficulty. We perform distillation via $50K$ randomly generated short audio clips from two (*i.e.* Ours$_A$ and Ours$_B$), 5mins YouTube videos (see Appendix A for more details). In Table 5, we compare the results against the performance of the teacher model and distilling directly using source dataset. We find that even for audio, distilling with merely a single audio clip's data provides enough supervisory signal to train the student model to reach above $80\%$ accuracy in the majority of the cases. In particular, we see significant improvement in distillation performance when a single audio has a wide variety of sounds to boost accuracy on challenging Voxforge dataset from $72.8\%$ to $78.4\%$. Our results on audio recognition tasks demonstrate the modality agnostic nature of our approach and further highlight the capability to perform knowledge distillation in the absence of large amount of data.

**Extension to video.** We conduct further experiments on video action recognition tasks. As distillation data, we generate simple pseudo-videos of 12 frames that show a linear interpolation of two crop locations at taken at the beginning and end for the City image and use X3D video archi-

Table 6: **Scaling to video.** Student models are trained from scratch. Top-1 accuracy is shown with 10-temporal center-cropped clips per video. Details in Appendix B.1

| Dataset | Teacher | **Distill** |
|---------|---------|---------|
| UCF-101 | 90.4 | 75.2 |
| K400 | 67.8 | 51.8 |

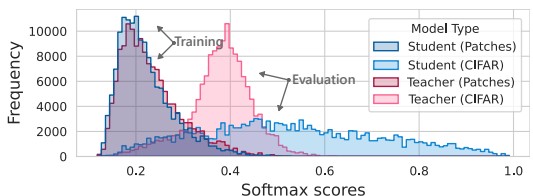

Figure 3: **Output confidence scores.** Temperature-scaled softmax scores of the predicted classes for 5K patches and the 5K CIFAR-10 validation set images.

Table 7: **Larger scale datasets.** Teacher is ResNet-18 and student is varied between ResNet-50 and ResNet-50x2. Students are trained from scratch. Gap to teacher in brackets for ResNet-50x2.

| | | | 1-Image-Distill | |
|---------|---------|---------|---------|---------|
| **Dataset** | **Categories (#Classes)** | **Teacher (R18)** | →R18 | →R50x2 |
| STL-10 (Coates et al., 2011) | Generic objects (10) | 96.3 | 93.9 | 95.0 (-1.3) |
| ImageNet-100 (Tian et al., 2020b) | Objects (100) | 89.6 | 84.4 | 88.5 (-1.1) |
| Flowers (Nilsback & Zisserman, 2006) | Flower types (102) | 87.9 | 81.5 | 83.8 (-4.1) |
| Places (Zhou et al., 2017) | Scenes (365) | 54.0 | 50.3 | 53.1 (-0.9) |
| ImageNet (Deng et al., 2009) | Objects (1000) | 69.5 | 66.2 | 69.0 (-0.5) |

tectures (Feichtenhofer, 2020) as teachers and students, see Appendix B.1 for further details. In Table 6, we find that our approach indeed generalizes well to distilling video models with performances of 75.2% and 51.8% on UCF-101 (Soomro et al., 2012) and K400 (Carreira & Zisserman, 2017)–despite the fact that the source data is not even a real video.

## 4.4 SCALABILITY TO LARGE-SCALE DATASETS AND MODELS

From this section on, we scale our experiments to larger models utilizing 224×224-sized images, and evaluate these on various vision datasets.

**Varying datasets.** In Table 7 we find that, overall, a single image is not enough to fully recover the performance on more difficult datasets. This might be because of the fine-grained nature of these datasets, *e.g.* ImageNet includes more than 120 kinds of dogs and Flowers contains 102 types of flowers. Nevertheless, we find a surprisingly high accuracy of 69% on ImageNet's validation set, even though this dataset comprises 1000 classes, and the student only having seen heavily augmented crops of a single image. In Table 8, we conduct further analyses on distilling ImageNet models.

**Patchification vs more data.** At the top part of the Table 8, we analyse the effect of using original vs patchified (p) random images from the training set of ImageNet. We find that while for 10 and 100 source images, patchification improves performance, this is not the case when a larger number of images is used. This indicates that increased diversity is especially a crucial component for the small data regime, while capturing larger parts of objects becomes important only beyond this stage. We also make the surprising observation that patches from a single high-quality image ("City") obtain better performances than patches generated from 1000 ImageNet training images. To understand why this might be the case, we note that the high coherency of training patches from a single image strikingly resembles what babies see during their early visual development, *e.g.*, looking at only few toys and people but from many angles (Bambach et al., 2018; Orhan et al., 2020). It is hypothesized that this unique combination of coherence and variability is ideal for learning to recognize objects (Orhan et al., 2020).

**Teacher/student architectures.** In Fig. 4, we analyze the performances of varying ResNet models by depth or width. We find that overall varying the width is a more parameter-efficient way to obtain higher performances, almost reaching the teacher's 69.5% performance with a ResNet-50x2 at 69.0%.

Table 8: **IN-1k distillations.** We vary distillation dataset and teacher/student configurations. '-' refers to experiments not continued due to compute cost outweighing insights. (p) means images are patchified. '†' refers to models trained with half the batch size due to memory constraints. 'AM' refers to only using the argmax as teaching signal.

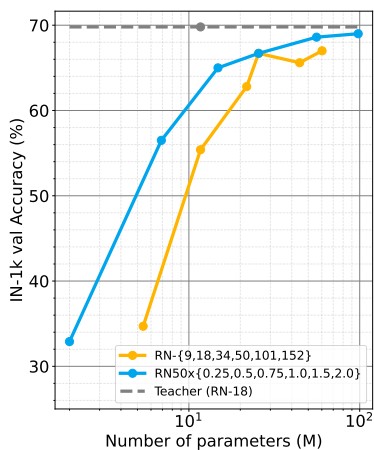

|     |           | **Setting** |            |   | **Epochs** |      |      |      |      |
|-----|-----------|-------------|------------|---|------|------|------|------|------|
|     | **Images** | Teacher    | Student    |   | 10   | 20   | 30   | 50   | 200  |
| (a) | 10        | R50         | → R50      |   | 7.6  | 13.3 | -    | -    | -    |
| (b) | 10 (p)    | R50         | → R50      |   | 17.2 | 27.3 | -    | -    | -    |
| (c) | 100       | R50         | → R50      |   | 14.7 | 25.1 | 32.3 |      |      |
| (d) | 100 (p)   | R50         | → R50      |   | 23.6 | 36.1 | 42.8 | -    | -    |
| (e) | 1000      | R50         | → R50      |   | 38.0 | 52.4 | 57.4 | 62.5 | -    |
| (f) | 1000 (p)  | R50         | → R50      |   | 27.1 | 39.4 | 45.2 | 50.9 | -    |
| (g) | Noise (p) | R18         | → R50      |   | 0.1  | 0.1  | 0.1  | -    | -    |
| (h) | Bridge (p)| R18         | → R50      |   | 21.2 | 34.8 | 40.0 | -    | -    |
| (i) | City (p)  | R18         | → R50      |   | 34.5 | 47.0 | 52.2 | 56.8 | 66.2 |
| (j) | City (p)  | R101        | →† R101    |   | 22.4 | 31.4 | -    | -    | -    |
| (k) | City (p)  | R50⋆        | → R50      |   | 4.6  | 6.6  | -    | -    | -    |
| (l) | City (p)  | R50         | → R50      |   | 18.0 | 34.0 | 39.8 | 45.6 | 55.5 |
| (m) | City (p)  | R18 (AM)    | →† R50x2   |   | 12.5 | 20.7 | 24.4 | 30.2 | 43.8 |
| (n) | City (p)  | R18         | →† R50x2   |   | 46.1 | 55.6 | 59.7 | 63.1 | 69.0 |

Figure 4: **Effect of student architecture.** We vary the architecture of the student by depth and width. Teacher is a ResNet-18, settings as in Table 7 with half batch size.

In addition, in the bottom half of Table 8 (rows *g-n*), we find surprising results: First, in contrast to normal KD, we find that the teacher's performance is not directly related to the final student performance, *e.g.* a ResNet-50 is less well-suited for distilling than a ResNet-18 (rows *i* vs *l*).

Second, we find that the choice of source image is even more important for ImageNet classification (see rows *g-i*), as switching from the "City" even to the "Bridge" one decreases performance considerably and the Noise image does not train at all. Third, we find that the best settings are those in which the student's capacity is higher than that of the teacher (rows *i,n* and Fig. 4). In fact this R18→R50x2 setup (row *n*) obtains a performance on ImageNet-12 of 69.0% which matches the teacher's performance within 0.5%. In Appendix B.3, we further compare distillations to ViT architectures (Dosovitskiy et al., 2021), but generally find lower performances (up to 64%), showing that the inductive biases of CNNs might positively aid in learning.

## 4.5 ANALYSIS

To better understand how our method is achieving these performances, we analyse the models.

**Output confidence scores.** In Fig. 3, we visualize the distribution of the confidence scores of the predicted classes for student and teacher models for the training and evaluation data. We observe that the student indeed learns to mimic the teacher's predictions well for the patch training data. However, during evaluation on CIFAR images, the student model exhibits a much broader distribution of values, some with extremely high confidence scores. This might indicate that the student has learned to extract meaningful discriminative features of object classes from the training patches, when compared to validation set inputs. For this, we fit a t-SNE (Van der Maaten & Hinton, 2008) using the features of 5K training patches and 5K test set images in Fig. 5. We find that, as expected, the training patches mapped to individual CIFAR classes do not resemble real counter parts. Moreover, we observe that all the training patches are embedded close to each other in the center, while CIFAR images are clustered towards the outside. This shows that the network is indeed learning features that are well suited for effective extrapolation.

**Neuron visualizations.** We next analyze a single-image distilled model trained to predict IN-1k classes. In Fig. 6, we visualize four final-layer neurons using the Lucid (Olah et al., 2017) library. When we compare neurons of the standard ImageNet supervised and our distilled model, we find that the neurons activate for very similar looking inputs. The clear neuron visualizations for "panda"

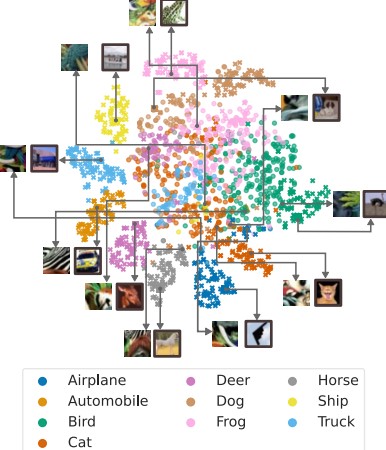

Figure 5: **Feature space visualization.** $t$-SNE of jointly embedding random patches (·) and CIFAR-10 test set (×) instances with our model distilled to classify semantic classes. We also show example images from patches and actual validation set images (enclosed in a box). Note how most training images are contained within a small region while the network needs to extrapolate for the real images which occupy the outer regions in this plot. Best viewed zoomed-in.

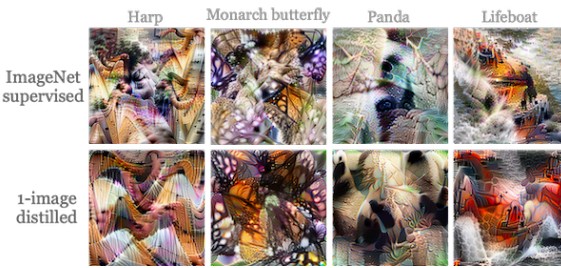

Figure 6: **Visualizing neurons** via activation maximisation of final layer neurons. We compare an ImageNet supervisedly trained model against our model which has been trained with single image distillation. Best viewed zoomed-in. We provide further examples in Appendix B.6.

or "lifeboat" (and more are provided in Appendix B.6) are especially surprising since this network was trained using only patches from the *City* Image, and has never seen any of these objects during its learning phase.

## 5 DISCUSSION

**The augmented image prior for extrapolation.** The results in this paper suggest a summary formulation as follows: *Within the space of all possible images $\mathcal{I}$, a single real image $x \in \mathcal{I}$ and its augmentations $\mathcal{A}(x)$ can provide sufficient diversity for extrapolating to semantic categories in real images.*

**Limitations.** In this paper, we were mainly concerned with showing that extrapolating from a single image works, empirically. Due to the limited nature of our compute resources, we have not exhaustively analyzed the choice of initial image or patchification augmentations, and longer training (as shown in (Beyer et al., 2022)) would further improve performances.

**Potential negative societal impact.** While our research question is of fundamental nature, one possible negative impact could be that the method is used to steal models and thus intellectual property from API providers – although in practice the performance varies especially for large-scale datasets (see Table 7).

**Conclusion and outlook.** In this work, we have analyzed whether it is possible to train neural networks to extrapolate to unseen semantic classes with the help of a supervisory signal provided by a pretrained teacher. Our quantitative and qualitative results demonstrate that our novel single-image knowledge distillation framework can indeed enable training networks from scratch to achieve high accuracies on several architectures, datasets and domains. This demonstrates that knowledge distillation can be done with just a single image plus augmentations, and also raises several further research questions, such as the dependency of the source image and the target semantic classes; how networks combine features for extrapolation; and the role and informational content of augmentations; all of which we hope inspires further research

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

# The augmented image prior: Distilling 1000 classes by extrapolating from a single image

# Appendix

# Table of Contents

## A   ADDITIONAL VISUALIZATIONS

### A.1   INPUT IMAGES

**Image sources and licences.**   From top to bottom, the images in Fig. 7 have the following licences and sources: (a): CC-BY (we created it), (b)[*] public domain (NASA) (c)[†] CC BY-SA 3.0 (by David Ball), (d)[‡] pixabay licence (personal and commercial use allowed), (e)[§] personal use licence.

Images (b), (d) and (e) are identical to the ones in the repo of (Asano et al., 2020), while (a) we have recreated on our own and (c) is a replacement for a similar one whose licence we could not retrieve.

---

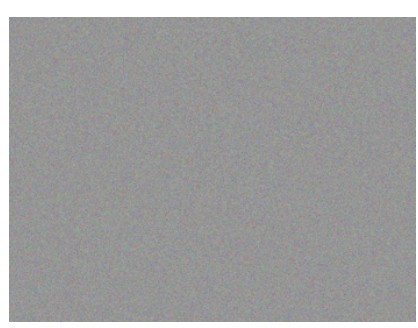

(a) The "Noise" Image. From uniform noise [0,255]. Size: 2,560x1,920, PNG: 16.3MB.

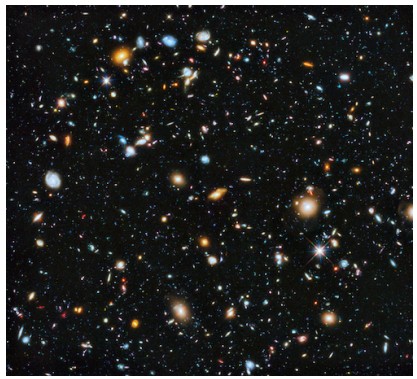

(b) The "Universe" Image. Size: 2,300x2,100, JPEG: 7.2MB.

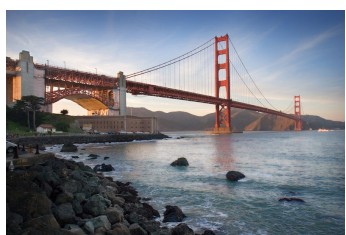

(c) The "Bridge" Image. Size: 1,280x853, JPEG: 288KB.

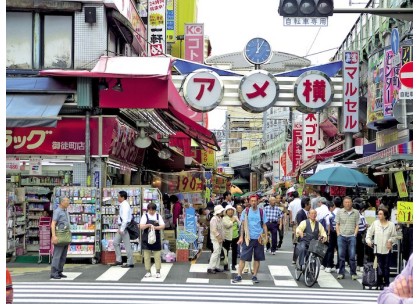

(d) The "City" Image. Size: 2,560x1,920, JPEG: 1.9MB.

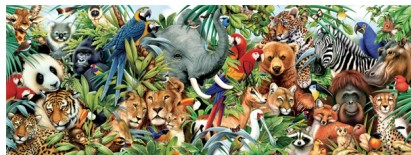

(e) The "Animals" Image. Size: 1,300x600, JPEG: 267KB.

Figure 7: **Single Images analysed.** Here we show the images analysed in Table 1.

**Audio source.** For the experiment on audio representation distillation, we use a 5.5mins English news-clip taken from BBC about how can Europe tackle climate change[¶] and a 5mins clip about 11 Germanic languages[‖].

## A.2 TRAINING PATCHES AND PREDICTIONS

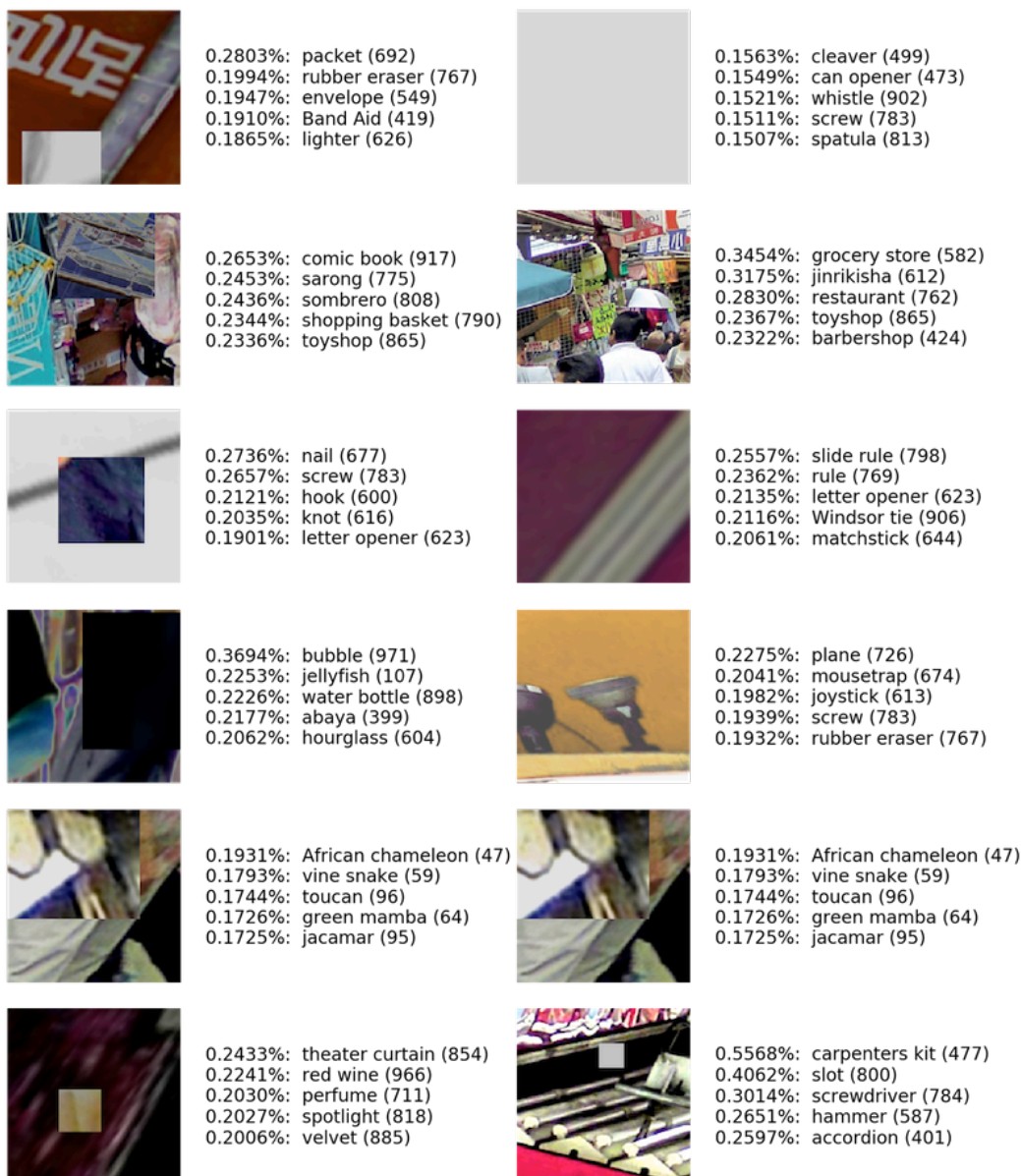

Figure 8: **ImageNet training data.** Here we visualize the augmented training data used for training along with the temperature-scaled Top-5 predictions of the ImageNet pretrained ResNet-50 teacher.

In Fig. 8, we show several training patches as they are being used during training along-side the teacher-network's temperature-adjusted predictions. Only very few training samples can be interpreted (*e.g.* 3rd row: "nail" or "slide rule"), while for most other patches, the teacher is being very "creative".

---

[¶] https://www.youtube.com/watch?v=nZgD4iPapVo
[‖] https://www.youtube.com/watch?v=iq2_gTETBXM

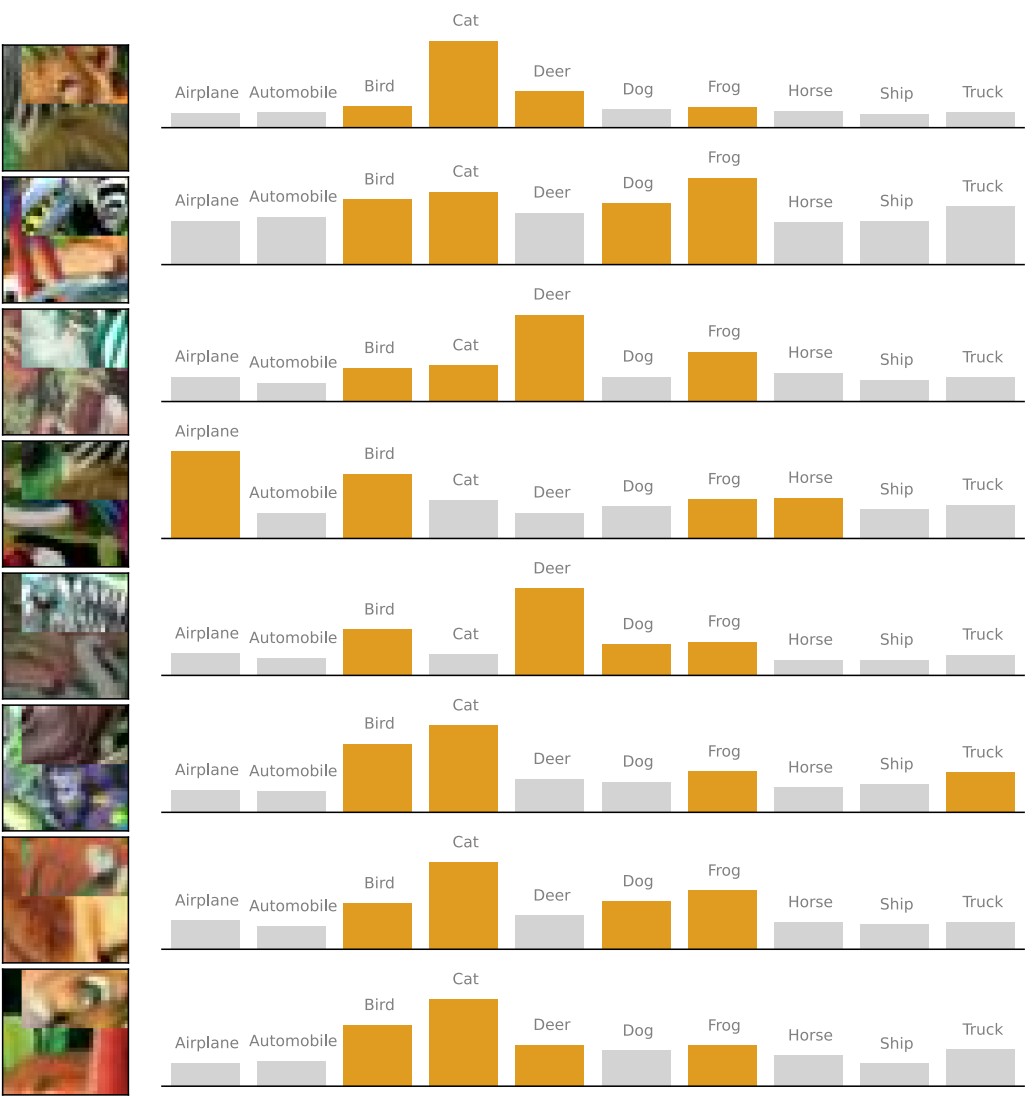

Figure 9: **CIFAR-10 training data.** We visualize temperature scaled predictions of WideResNet-40-4 teacher on random patches generated from the Animal image (see Figure 7).

In Fig. 9, we show a similar plot for the training patches used in CIFAR-10 training. Here we show the *whole* teaching signal, and highlight the top-5 predictions of the teacher for visualization.

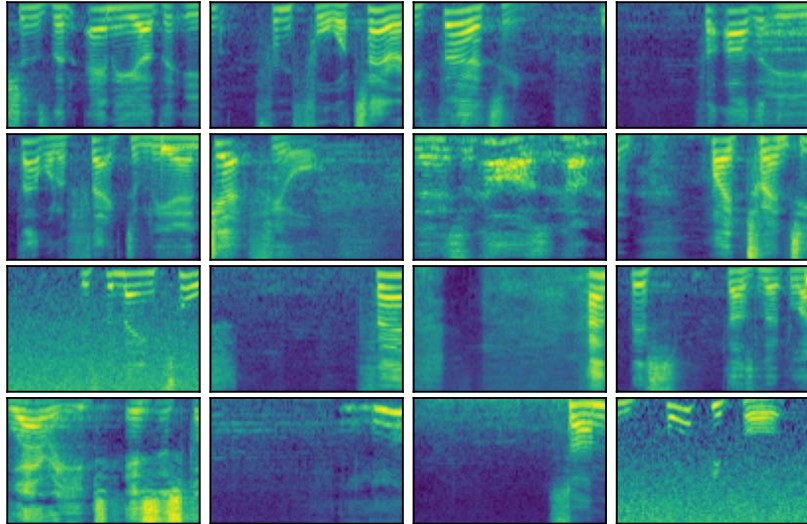

Figure 10: **Audio spectrograms.** We visualize spectrograms generated from a one second segment of the augmented audio. We generate 50K audio clips from a 5 minutes audio recording by randomly taking a cropped audio segment of 2 seconds and augmenting via one of the transformation functions mentioned in Section C.1. During model training, we use 1 second segment and apply MixUp augmentation that varies across batch.

Finally, in Fig. 10, we show some some augmented training samples for the audio classification experiment. This figure shows how various spectrograms can be generated from a single clip by utilizing many augmentations.

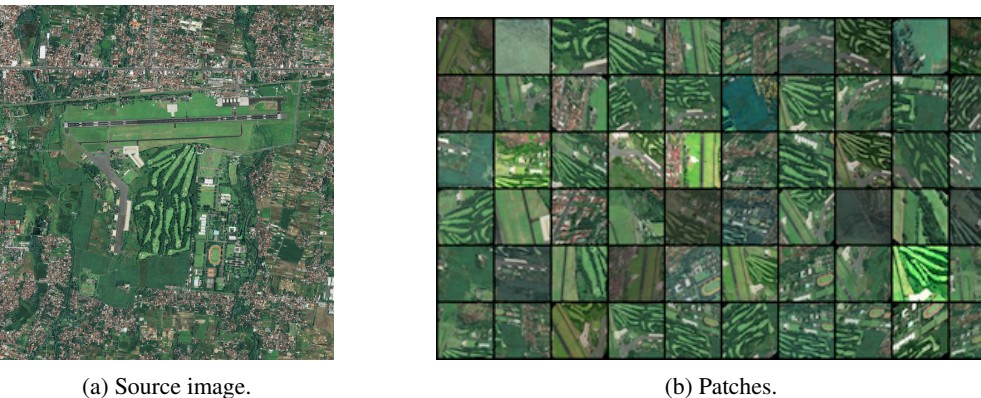

(a) Source image.                                     (b) Patches.

Figure 11: **Satellite image patches.** We visualize random patches of size $32 \times 32$ generated from a single satellite image of Yogyakarta Airport, Indonesia. The image is of size $3846 \times 4087$ and 17MB large as a JPEG.

## B  ADDITIONAL EXPERIMENTS

### B.1  VIDEO ACTION RECOGNITION

We conduct further experiments on video action recognition tasks. For this we use the common UCF-101 (Soomro et al., 2012) dataset, more specifically the first split. As distillation data, we

generate simple source videos of 12 frames that show a linear interpolation between two crops of the City image. We generate 200K of these videos and apply AugMix (Hendrycks et al., 2019) and CutMix (Yun et al., 2019) during the distillation for additional augmentations. For the architectures we use the recent, state-of-the-art X3D architectures (Feichtenhofer, 2020) as they obtain strong performances and allow for flexible scaling of network architectures. We use a X3D-XS model as the teacher, which is trained with inputs of $160 \times 160$ and 4 frames and a temporal subsampling factor of 12. For the student model we scale this X3D-XS model in terms of width and depth by increasing the depth factor and width factor from 2.2 to 3 and 4 respectively, yielding a network with 18.1M parameters compared to 3.2M. For UCF-101 we finetune the teacher model starting from supervised Kinetics-400 (Carreira & Zisserman, 2017) pretraining, which achieves a 90.4% performance. We use a batch size of 128 per GPU on two GPUs, a learning rate of $10^{-3}$, a temperature of 5, cosine learning rate schedule with a warmup of 5 epochs, with no weight decay and no dropout. As this experiment takes even longer than the image ones, we have not conducted systematic ablations on these parameters. Instead, we merely picked ones which looked promising after a few epochs but believe that even this small experimental evidence is enough to investigate to what extent these 1-image "fake-videos" can be used for learning to extrapolate.

The results are given in Table 6. We observe a performance of over 72% on UCF-101 using just a single fake-video as the training data along with the pretrained teacher. While this number is far below the state of the art or the teacher's performance, it for example outperforms CLIP's 69.8% zero-shot performance with a ViT-B16 model and shows that the student is able to extrapolate to the action classes of UCF from just one datum.

## B.2 VARYING DISTILLATION LOSS FUNCTIONS

In Table 9 we compare the standard KD loss against L1 and L2 losses (using scaled logits as in the KD loss) for training the student. We find that while for CIFAR-10 the choice of the distillation method does not impact the performance, on the more difficult ImageNet dataset, there is a small difference of around $-0.4\%$ and $-0.5\%$ for L1 and L2 losses as compared to the KD loss at epoch 30. This shows firstly that our method is not dependend on a specific loss for distilling knowledge from a teacher to a student. Secondly, it shows that the standard KD loss is actually a strong baseline, as was also reported in (Tian et al., 2020a).

Table 9: **Distillation Loss Function Comparison.** We compare the distillation performance with different loss functions along with teacher and student supervised training. Results are shown for CIFAR-10 with a WiderRNet40-4→WiderRNet16-4 setting and for ImageNet with R18→R50. For ImageNet we show the performance progression after 10/20/30 epochs

| Data | Teacher | Student | L1 | L2 | KD |
|------|---------|---------|------|------|------|
| C10 | 95.4 | 95.2 | 93.3 | 93.4 | 93.4 |
| IN-1k | 69.8 | 76.1 | 36.2/45.1/51.8 | 25.1/45.9/51.7 | 34.5/47.0/52.2 |

## B.3 IMAGENET VISION TRANSFORMER DISTILLATIONS

In Table 10 we compare the best results from the main paper of distilling to ResNets to distilling to the ViT (Dosovitskiy et al., 2021) and CaiT (Touvron et al., 2021) architectures. We observe that while convolutional networks learn significantly faster, with *e.g.* differences of more than 30% at the tenth epoch. While the ViT architecture almost catches up at epoch 200 with a final performance of 63.9%, the CaiT model remains fairly low at 58.5%. Nevertheless, this experiments shows that distilling across architecture types works and that even less constrained Vision Transformers can be trained with our method simply.

## B.4 TRAINING WITH VARIOUS RANDOM NOISES

In Table 11, we experiment with adding various types of noise structures as inputs to the teacher and student instead of augmented patches from the images. Notably, we either input random uniform noise between [0,1] (row ($w$)), random normal noise with a mean of 0 and standard deviation of 1

Table 10: **Training Vision Transformer student models.** Setting as in Table 7 (from which first row is copied for comparison).

| Setting | | | Epochs | | | | |
|---|---|---|---|---|---|---|---|
| Image | Teacher | | Student | 10 | 20 | 30 | 50 | 200 |
| 1x City | R18 | $\to^{\dagger}$ | R50x2 | 46.1 | 55.6 | 59.7 | 63.1 | 69.0 |
| 1x City | R18 | $\to^{\dagger}$ | CaiT-S24 | 9.2 | 25.9 | 34.0 | 42.5 | 58.5 |
| 1x City | R18 | $\to^{\dagger}$ | ViT-B | 15.7 | 31.9 | 39.3 | 47.2 | 63.9 |

(row ($x$)) and convex combinations of the augmented input patches with the normal noise. We find that just like the augmented patches from a random noise image (row ($g$)), random noise as input also does not work for distilling semantic classes. While generating new random noise at every iteration does create more variability, in practice, a random noise image with augmentations likely already achieve a high amount of variance, showing that this is not the crucial component for learning, but instead having structures from a real image. This is further confirmed by the steady increases in performance when moving from row ($x$) to ($z$).

Table 11: **Training with various random noise.** Setting as in Table 7 (from which first 3 rows are copied for comparison).

| | Input | Setting | | Epochs | | | | |
|---|---|---|---|---|---|---|---|---|
| | | Teacher | Student | 10 | 20 | 30 | 50 | 200 |
| (g) | Noise Image (p) | R18 | $\to$ R50 | 0.1 | 0.1 | 0.1 | - | - |
| (h) | Bridge Image (p) | R18 | $\to$ R50 | 21.2 | 34.8 | 40.0 | - | - |
| (i) | City Image (p) | R18 | $\to$ R50 | 34.5 | 47.0 | 52.2 | 56.8 | 66.2 |
| (v) | StyleGAN Baradad et al. (2021) | R18 | $\to$ R50 | 15.4 | 30.1 | 37.8 | 44.4 | 60.4 |
| (w) | Random Uniform [0,1] Noise | R18 | $\to$ R50 | 0.1 | 0.1 | 0.1 | 0.1 | - |
| (x) | Random Normal (0,1) Noise | R18 | $\to$ R50 | 0.1 | 0.1 | 0.2 | 0.2 | - |
| (y) | 0.8x Random Normal (0,1) + 0.2x City (p) | R18 | $\to$ R50 | 0.2 | 0.3 | 0.4 | 0.7 | - |
| (z) | 0.5x Random Normal (0,1) + 0.5x City (p) | R18 | $\to$ R50 | 7.6 | 13.1 | 17.1 | 22.3 | - |

## B.5 STANDARD DEVIATIONS IN PERFORMANCE

In Table 12, we report the performance of five independent runs on CIFAR-10 dataset to further highlight the robustness of our experimental results. We notice consistent accuracy scores across different training runs with minor changes of $\approx \pm 0.2$.

Table 12: Model training with different random seeds. We compare the performance of five independent runs on CIFAR-10 dataset. We use WRN40-4 as teacher and WRN16-4 as student. "Full" shows the accuracy distilling using CIFAR-10 data, and "Ours" represents distillation using random patches generated from "Animals" image.

| Teacher | Student | Full | Ours |
|---|---|---|---|
| $95.38 \pm 0.09$ | $94.76 \pm 0.22$ | $93.26 \pm 0.17$ | $93.38 \pm 0.07$ |

## B.6 NEURON ACTIVATION MAXIMIZATIONS

In Fig. 12, we show further class-neuron activation maximization visualizations for the R50→R50 distillation experiment and compare them to the corresponding ones of the supervised teacher model. Similar to the examples in the main paper, we find that the neurons visualizations are remarkably similar.

In Fig. 13, we show further visualizations of the trained student model from the R50→R50 distillation experiment. We show five randomly chosen neurons for every layer in the ResNet.

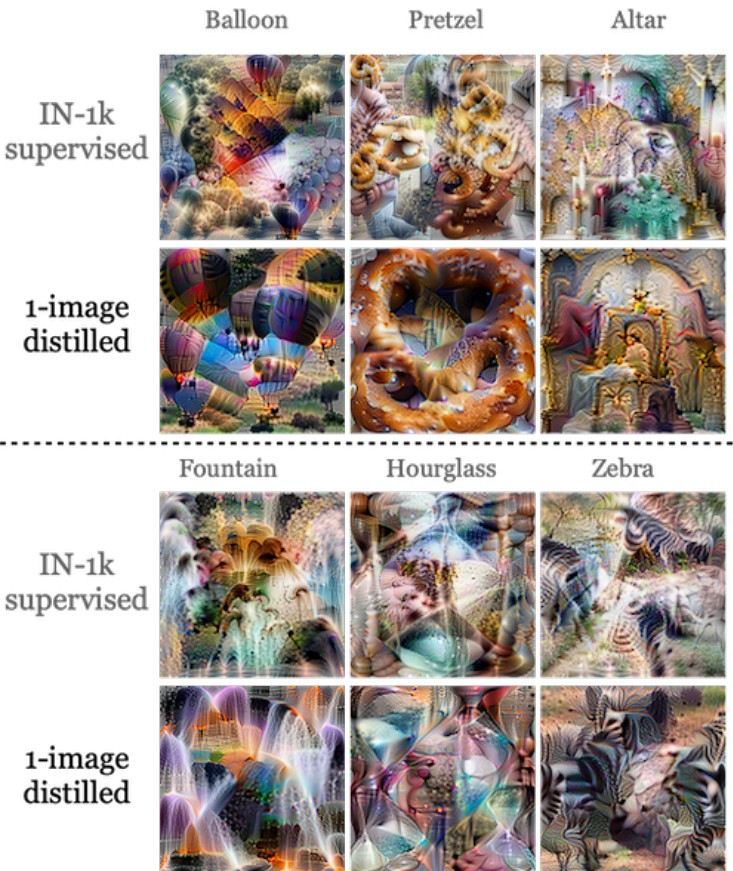

Figure 12: **Visualizing neurons** via activation maximisation of final layer neurons. We compare an ImageNet supervisedly trained model against our model which has been trained with single image distillation.

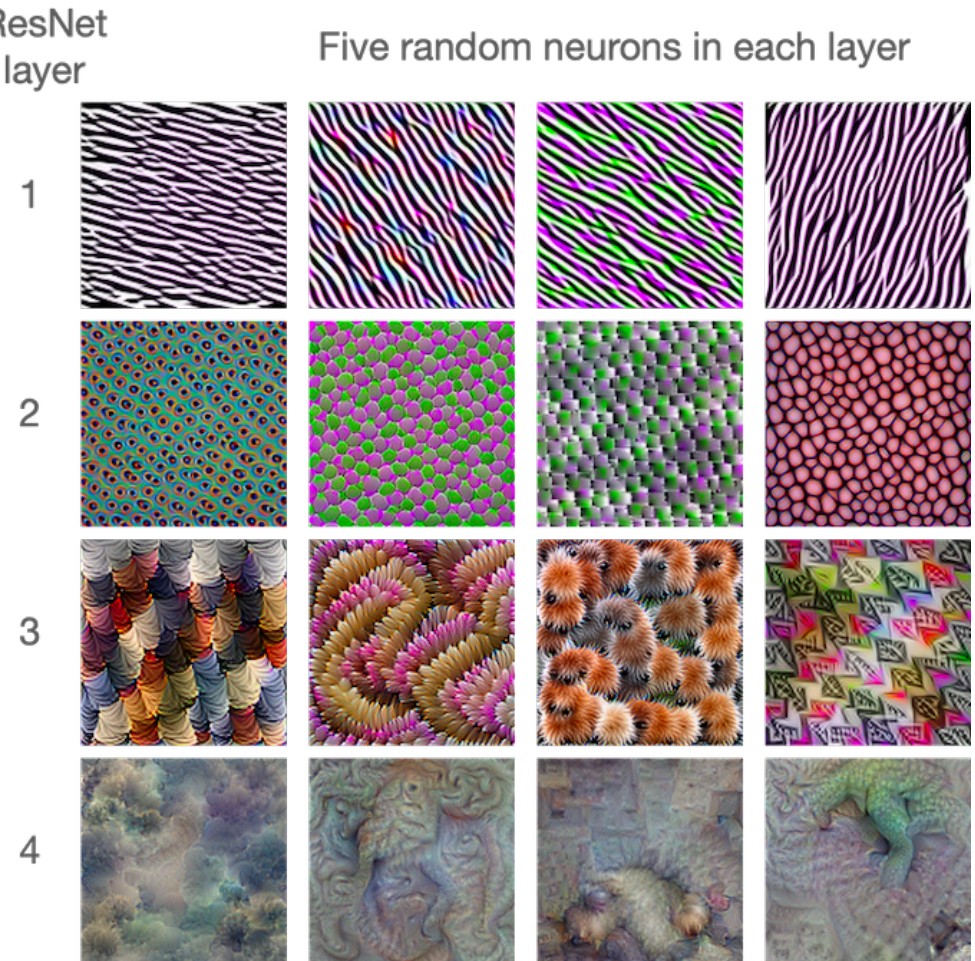

Figure 13: **Visualizing intermediate neurons** via activation maximisation. We visualize five randomly picked neurons for each layer and show their maximally responding input images.

## B.7 T-SNE OF FEATURE SPACE

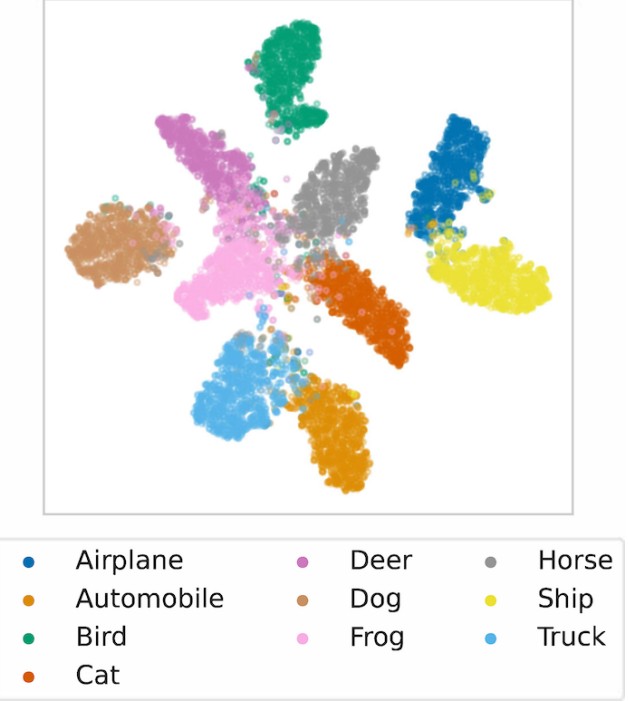

Figure 14: **t-SNE embeddings of CIFAR-10**. We visualize t-SNE embeddings of a WideR16-4 model distilled from WideR40-4 with random patches generated from an 'Animal' image.

We visualize representations of distilled student model on CIFAR-10 validation set in order to highlight semantic relevance of the features in Fig. 14. We extract the feature from penultimate layer of WideR16-4 model for computing 2-D t-SNE embeddings. The visualization clearly highlights that distilled features contain substantial amount of semantic information to correctly differentiate between different object categories.

## B.8 CENTERED KERNEL ALIGNMENT COMPARISON

In Fig. 15, we visualize similarity between different layers of the models trained in a general supervised manner and with distillation using patches of 1-image for CIFAR-10 and ImageNet datasets. We use centered kernel alignment (Kornblith et al., 2019) method to identify the resemblance of features learned with these different training approaches. We notice high similarity in representations, which reinforce our empirical evaluation that the student models indeed learn useful features required for differentiating between semantic categories.

## B.9 SINGLE-IMAGE TRAINING DATA VS TODDLER DATA

In Fig. 16a, we compute the distances of GIST features (Oliva & Torralba, 2001) of 10K training images of the "City" image at resolution $256 \times 256$ in Fig. 16a. Following (Bambach et al., 2018), we L2 normalize these GIST features before computing pair-wise distances and plotting the histogram of the values. To make the comparison of our training data with the visual inputs of toddlers more concrete, we compare low-level GIST (Oliva & Torralba, 2001) features of our 1-image dataset against those computed from visual inputs of a toddler in (Bambach et al., 2018). As we do not have access to the dataset of (Bambach et al., 2018), we copy their Figure 3b for reference in Fig. 16b. While the GIST distances for ImageNet are very different with a mean around $0.75$ (Bambach et al., 2018), we find that our single image dataset's distribution in Fig. 16a closely resembles that of the visual inputs of a toddler. While this is one possible explanation for why this type of data might work well for developing visual representations, further research is still required.

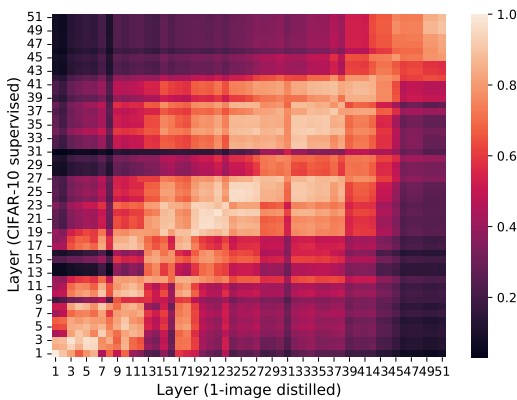

(a) WideR16 − 4 via CIFAR-10 supervised training vs 1-image distilled.

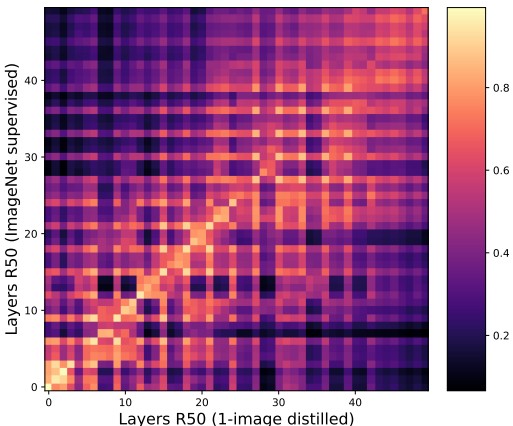

(b) R50 via ImageNet supervised training vs 1-image distilled.

Figure 15: **Representation similarity with CKA Kornblith et al. (2019).** We visualize the similarity between supervised representations and those distilled using random patches of 1−image. As in Nguyen et al. (2020), we compute CKA for all the layers in a model using CIFAR-10 testset. The layers in the same block group shows high similarity though both models are trained in different ways, indicating that distilled features are highly similar to those learned in a supervised way.

## B.10 DATA-FREE PRUNING AND QUANTIZATION OF PRE-TRAINED MODELS

Neural network compression has been studied extensively in the literature to produce light-weight models with the objective of improving computational efficiency. In addition to knowledge distillation, network pruning and quantization are other well-known approaches. In the former, the goal is to remove or sparsify (zeroing-out) the network's weights based on a specific criteria, such as subset of weights with the lowest magnitude. The later is concerned with reducing precision from 32-bits floating numbers to 8-bits and even single bit as binary neural networks. In general, from practical standpoint post-training compression is largely employed in conjunction with fine-tuning to avoid accuracy drop.

Here, we study utilize our proposed single-image distillation framework for post-training model compression without using any real data. Specifically, we ask the question, whether a pre-trained model can be compressed when no in-domain data is available? To this end, we utilize Tensorflow Model Optimization Toolkit** to prune and quantize various pre-trained model on CIFAR-10 and

---

**https://www.tensorflow.org/model_optimization

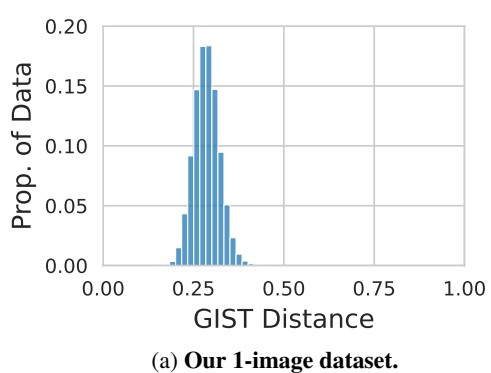

(a) **Our 1-image dataset.**

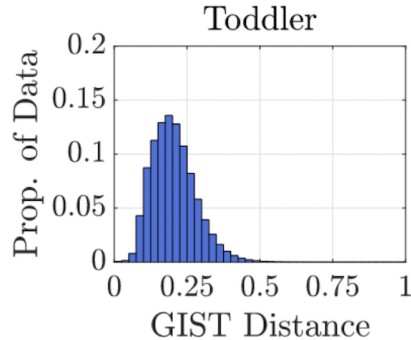

(b) **Toddler data.** Figure from (Bambach et al., 2018).

Figure 16: **Pair-wise distances of L2-normalized GIST features** for 10K images from our 1-image training dataset used for our ImageNet experiments

| Dataset | Model | Standard | Quantization *(source)* | Ours - Quantization *(single-image)* | Pruning *(source)* | Ours - Pruning *(single-image)* |
|---|---|---|---|---|---|---|
| CIFAR-10 | ResNet56 | 93.77 | 93.64 | 93.45 | 93.38 | 93.18 |
| | VGG11 | 91.57 | 91.22 | 90.97 | 91.14 | 90.86 |
| | VGG19 | 93.28 | 92.94 | 93.00 | 92.84 | 92.96 |
| | WideRNet16-4 | 94.81 | 94.40 | 94.59 | 94.76 | 94.43 |
| | WideRNet40-4 | 95.42 | 95.09 | 94.90 | 95.29 | 94.82 |
| CIFAR-100 | ResNet56 | 70.99 | 70.89 | 69.85 | 70.74 | 70.42 |
| | VGG11 | 69.65 | 69.92 | 68.86 | 69.77 | 69.13 |
| | VGG19 | 70.79 | 70.60 | 70.21 | 70.89 | 70.56 |
| | WideRNet16-4 | 75.81 | 75.45 | 74.71 | 75.68 | 75.51 |
| | WideRNet40-4 | 78.14 | 78.27 | 77.60 | 77.94 | 77.78 |

Table 13: **Self-distillation with a single-image for efficient deep models**. We perform model compression via self-distillation using source data and 50k random patches generated from 'Animals' image for 50% sparsity (in case of pruning) and 8-bits quantization without any noticeable loss in performance.

CIFAR-100 datasets. We use self-distillation, where a pre-trained model acts as teacher and student clone of it is compressed during fine-tuning phase. In Table 13, we report accuracy scores for the pruned and quantized models in comparison with the standard models and those compressed using real in-domain data. In all the cases, we can notice that there is negligible loss in accuracy, i.e., $<=$ 1%, while using random patches.

| Sparsity | ResNet56 | | VGG11 | | WideRNet16-4 | |
|---|---|---|---|---|---|---|
| | C10 | C100 | C10 | C100 | C10 | C100 |
| 0% | 93.77 | 70.99 | 91.57 | 69.65 | 94.81 | 75.81 |
| 25% | 93.55 | 70.85 | 91.03 | 69.37 | 94.55 | 75.40 |
| 50% | 93.18 | 70.74 | 90.86 | 69.13 | 94.43 | 75.51 |
| 75% | 92.68 | 67.87 | 90.54 | 67.81 | 94.10 | 74.05 |
| 85% | 91.50 | 61.22 | 89.12 | 65.23 | 93.05 | 71.46 |

Table 14: **Varying Sparsity**. Pruning results achieved on CIFAR-10/100 for different sparsity rates using self-distillation with random patches generated from the 'Animal' image.

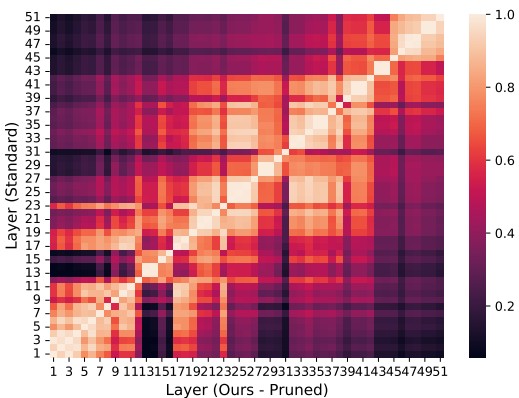

Figure 17: **Representation similarity with CKA of a pruned and standard WideR16-4 models trained on CIFAR-100.**

Further, in Table 14, we vary sparsity ratio and can observe that a model can be pruned up to 75% sparsity with three-points drop in accuracy but with 25% pruning ratio the loss is merely notice-able. In Figure 17, we analyze representational similarity of pruned WideRNet16-4 model with the original trained on CIFAR-100, the diagonal entries highlight that even after 50% sparsity the representations are very similar. To summarize, with model compression use case, we barely scratched the surface of what is possible with our single-image distillation framework and we hope it will inspire further studies and applications in other domains.

### B.11 TRANSFER LEARNING EXPERIMENTS

In this section, we compare the performances on downstream tasks of our teacher network (ResNet-18), our single-image distilled network (ResNet-50) and another ResNet-50, purely trained in a self-supervised manner on our augmented single-image dataset. For this, we pretrain using the official code of MoCo-v2 (He et al., 2020) in the standard 200 epoch setting, while applying their default "v2" augmentations during training.

**Evaluation.** We follow standard linear evaluation procedure from the MoCo-v2 repo. which uses a batch size of 256, learning rate of 30 which is multiplied by 0.1 at epochs 60 and 80 for a total of 90 epochs. For the data-efficient full-fine tuning, we utilize the implementation from SwAV (Caron et al., 2019), which trains for 20 epochs using a cosine decay learning rate schedule and a batch size of 256. For the data-efficient SVM classification experiments, we follow the implementation of (Li et al., 2021) and report average results from 5 trials and keep the SVM's cost parameter fixed at a value of 0.5. For the COCO object detection experiments we use the detectron2 repo (Wu et al.,

Table 15: **Representation learning performance.** We compare our distillation approach (R18→R50) against pretraining via MoCo-v2 on the same data. We report: linear eval. accuracy on IN-1k and Places; accuracy when finetuning and using 1% of ImageNet with labels; SVM classification mAP on PVOC07 and accuracy on Places with varying numbers of images per class; mAP for COCO-detection with FPN.

| | IN-1k | Places | IN-1k (1%) | PVOC | | Places | | COCO |
| imgs / class | ~1K | ~10K | ~13 | 4 | 16 | 4 | 16 | NA |
|---|---|---|---|---|---|---|---|---|
| IN-1k teacher | [69.8] | 44.1 | [69.8] | 65.7 | 77.1 | 21.4 | 30.4 | - |
| IN-1k R50 | [76.2] | 51.5 | [76.2] | 73.8 | 82.3 | 27.0 | 35.4 | 38.9 |
| 1-image **MoCo-v2** | 28.5 | 28.8 | 14.4 | 17.2 | 26.6 | 4.5 | 9.7 | 35.6 |
| 1-image **Distill** | 68.8 | 47.2 | 64.1 | 58.9 | 73.5 | 21.0 | 31.3 | 35.4 |

| | | | Noise | | | Blur | | | | Weather | | | | Digital | | | |
| Network | Error | mCE | Gauss. | Shot | Impulse | Defocus | Glass | Motion | Zoom | Snow | Frost | Fog | Bright | Contrast | Elastic | Pixel | JPEG |
|---|---|---|---|---|---|---|---|---|---|---|---|---|---|---|---|---|---|
| IN-1k R18 | 30.2 | 84.7 | 87 | 88 | 91 | 84 | 91 | 87 | 89 | 86 | 84 | 78 | 69 | 78 | 90 | 80 | 85 |
| *students* | | | | | | | | | | | | | | | | | |
| 1-image R50x2 | 31.0 | 85.9 | 88 | 89 | 91 | 85 | 92 | 88 | 89 | 88 | 86 | 82 | 71 | 80 | 91 | 82 | 87 |
| 1-image R50 | 33.8 | 89.8 | 93 | 93 | 96 | 87 | 94 | 90 | 92 | 91 | 90 | 84 | 77 | 83 | 96 | 87 | 93 |

Table 16: **IMAGENET-C evaluations.** Clean Error, mCE, and Corruption Error values of different corruptions and architectures. The mCE value is the mean Corruption Error of the corruptions in Noise, Blur, Weather, and Digital columns.

2019) and the *1x* schedule using a FPN (Lin et al., 2017) as in (He et al., 2020). We additionally vary the learning rates from 1x and 4x as the models are trained with very different losses compared to the supervised variant for which the learning schedule is made. The results are shown in the last row of Table 15 and show that our models benefit from a higher initial learning rate, which also does not have any negative effect on training longer *e.g.* with the 2x schedule, while the MoCo pretrained model works best with the 1x learning rate.

We find that freezing the backbone and only training a linear layer on top of it, can achieve performances of 68.7% and 47.2% for ImageNet and Places, respectively, vastly outperforming the self-supervised variant. A similar trend is observed for data-efficient classification reaching close but lower performances to the teacher model.

## B.12 ROBUSTNESS EXPERIMENTS

In Table 16, we report results on ImageNet-C (Hendrycks & Dietterich, 2019).

## B.13 ANALYSIS OF PER-CLASS ACCURACIES

**Worst underperforming 10 classes.** The 10 classes that have the highest top-1 accuracy difference compared to the teacher model (for the R18→R50 setting) are: `Tibetan terrier, Lakeland terrier, golden retriever, dhole, Border terrier, cocker spaniel, Welsh springer spaniel, Brabancon griffon, collie, mountain bike`, with accuracies differences ranging from -34% to -20%. From the underperforming classes, we find that 9/10 of these are dog breeds. This illustrates how out single-image trained model lacks behind in very fine-grained classification compared to the ImageNet trained model. This is likely because the single-image lacks the necessary structures and patterns for disambiguating these classes despite the help of augmentations.

In Fig. 18, we plot the validation performance against the frequency of how often the class appears as a top-1 prediction of the teacher during 1 epoch of training and find that it is unrelated, showing the student is profiting heavily from the knowledge contained in the soft-predictions, echoing findings from (Hinton et al., 2015; Furlanello et al., 2018). We further compare these per-class performances

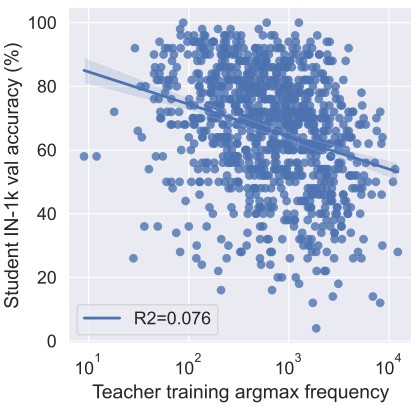
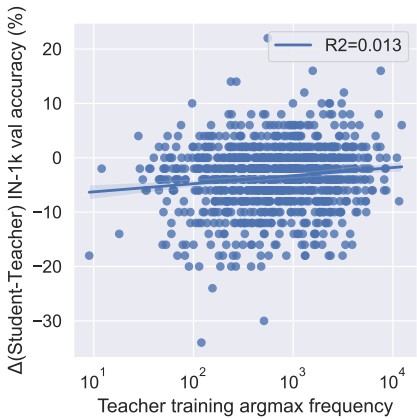

Figure 18: **Comparison of per-class accuracy.** Per-class student performance vs frequency of the class being the top-1 prediction of the teacher.

Figure 19: **Comparison of per-class relative accuracy.** Per-class performance relative to the teacher is plotted vs frequency of the class being the top-1 prediction of the teacher.

against the teacher's performance in Fig. 19 and find that there is no relationship between when the student under or overperforms the teacher vs how often a particular likeness of a class appears.

## C   IMPLEMENTATION DETAILS

All code will be released open-source and is attached as supplementary information to this submission.

### C.1   INITIAL PATCH GENERATION

We do not tune the individual patch generation and instead adapt the procedure directly from (Asano et al., 2020)[††]. The augmentations applied to the input image of size $H \times W$ to arrive at individual patches of size $P \times P$ is (in order) as follows:

1) `RC(size=0.5*min(H,W))`

2) `RRC(size=(1.42*P), scale=(2e-3, 1))`

3) `RandomAffine(degrees=30, shear=30)`

4) `RandomVFlip(p=0.5)`

5) `RandomHFlip(p=0.5)`

6) `CenterCrop(size=(P,P))`

7) `ColorJitter(0.4,0.4,0.4,0.1, p=0.5)`

All transformations are standard operations in PyTorch: RC stands for RandomCrop, *i.e.* taking a random crop in the image with a specific size; RRC for RandomResizedCrop, *i.e.* taking a random sized crop withing the size specified in the scale tuple (relative to the input); RandomAffine for random affine transformations (rotation and shear); RandomVFlip and RandomHFlip for random flipping operations in the vertical and horizontal direction, and CenterCrop crops the image in the center to a square image of size $P \times P$. Finally, ColorJitter computes photometric jittering, where the parameters for the strengths are given in the order of brightness, contrast, saturation and hue and applied with a certain probability.

---

[††]https://github.com/yukimasano/single_img_pretraining

Similarly, for audio clips generation given a single audio, we use augmentation operations from (Bitton & Papakipos, 2021) with default settings. Specifically, to create a single example we apply the following procedure: we randomly crop a segment of 2-seconds, and use randomly sample an augmentation function to create transformed instances and save them in mono format. In our work, we use these augmentations, all with their default settings:

1) `add-background-noise`

2) `change-volume`

3) `clicks`

4) `clip`

5) `harmonic`

6) `high-pass-filter`

7) `low-pass-filter`

8) `normalize`

9) `peaking-equalizer`

10) `percussive`

11) `pitch-shift`

12) `reverb`

13) `speed`

14) `time-stretch`

## C.2 Computing log-Mel spectrograms

The log-Mel spectrograms are generated on-the-fly during training from a randomly selected 1-second crop of an audio waveform as the model's input. We compute it by applying a short-time Fourier transform with a window size of 25ms and a hop size equal to 10ms to extract 64 Mel-spaced frequency bins for each window. During evaluation, we average over the predictions of non-overlapping segments of an entire audio clip.

## C.3 Audio neural network architecture

Our audio convolutional neural network is inspired by (Tagliasacchi et al., 2019) and it consists of four blocks. We perform separate convolutions in each block with a kernel size of $4$. One on the temporal and another on the frequency dimension, we concatenate their outputs afterward to perform a joint $1 \times 1$ convolution. It allows model to capture fine-grained features from each dimension and discover high-level features from shared output. We apply L2 regularization with a rate of $0.0001$ in each convolution layer and also use group normalization (Wu & He, 2018). Between the blocks, we utilize max-pooling to reduce the time-frequency dimensions by a factor of 2 and use a spatial dropout rate with a rate of $0.2$ to avoid over-fitting. We apply ReLU as a non-linear activation function and feature maps in the convolutions blocks are $24, 32, 64$, and $128$. Finally, we aggregate the feature with a global max pooling layer which are fed into a fully-connected layer with number of units equivalent to the number of classes.

## C.4 Training

### C.4.1 Optimization.

For each of these datasets, we first train a teacher network in a usual supervised manner, that is then used for the distillation. For the distillation to the student model, we use a temperature $\tau = 8$ motivated by findings in (Beyer et al., 2022). We keep this temperature fixed throughout the whole of the paper due to limited compute. For optimization, we use AdamW (Loshchilov & Hutter, 2018).

### C.4.2 SMALL-SCALE EXPERIMENTS

For experiments on CIFAR-10, CIFAR-100, and other smaller datasets, we use Tensorflow for running experiments on a single T4 GPU with a batch size of 512 using an Adam (Kingma & Ba, 2015) optimizer with a fixed learning rate of 0.001. We use standard augmentations including, random left right flip, and random crop. The supervised models also uses *cutout* augmentations with a cutout size of $16 \times 16$. For Mix-up, we sample $\alpha$ uniformly at random between zero and one. In Cut-Mix, we use a fixed value of 0.25 for $\alpha$ and $\beta$. With the following setup, each of the single image distillation experiment of 1K epochs took around $2 - 3$ days. The supervised and standard (using source data) distillation models are trained for 100 epochs (per epoch 2000 steps) with a batch size of 128 using SGD for optimization. For VGG (Simonyan & Zisserman, 2014) and ResNet (He et al., 2016) models, we use a learning rate schedule of $0.01, 0.1, 0.01, 0.001$ decayed at following steps $400, 32000, 48000, 64000$ with momentum of 0.9. For WideResNet (Zagoruyko & Komodakis, 2016a), we use a learning rate schedule of $0.1, 0.02, 0.004, 0.0008$ that is decayed at following steps $24000, 48000, 64000, 80000$ with Nesterov enabled.

In audio experiments, we use an Adam (Kingma & Ba, 2015) optimizer with a fixed learning rate of 0.001 and use batch size of 128 and 512 for standard models and single-clip distillation, respectively. Furthermore, we also utilize Mix-Up augmentation during our knowledge distillation experiments.

### C.4.3 LARGE-SCALE EXPERIMENTS

We use PyTorch's `DistributedDataParallel` engine for running experiments on 2 A6000 GPUs in parallel with batch-sizes of 512 each. For optimization we use AdamW (Loshchilov & Hutter, 2018) with a learning rate of 0.01 and a weight-decay of $10^{-4}$. These values were determined by eyeballing the results from (Beyer et al., 2022) to find a setting that might generalize across datasets, as we do not have enough compute to run hyper-parameter sweeps. With this setup, each 200 epoch experiment took around 5 days. We have also found that halving the batch-size performs equally well, and is more amenable to lower-memory GPUs. For the distillation experiments, we use Cut-Mix (Yun et al., 2019) with its default parameters of $\alpha = \beta = 1.0$.

### C.5 GIST FEATURES COMPARISON

In Fig. 16a, we compute the distances of GIST features (Oliva & Torralba, 2001) of 10K training images of the "City" image at resolution $256 \times 256$ in Fig. 16a. Following (Bambach et al., 2018), we L2 normalize these GIST features before computing pair-wise distances and plotting the histogram of the values.

