# OpenReview forum: "The Augmented Image Prior: Distilling 1000 Classes by Extrapolating from a Single Image"
_ICLR.cc/2023/Conference — ICLR 2023 poster_

### Official Review · Reviewer_AYLh · 2022-10-22

**Confidence:** 4
**Clarity, Quality, Novelty And Reproducibility:** The paper is well written. Details ar…
**Correctness:** 3
**Technical Novelty And Significance:** 3
**Empirical Novelty And Significance:** 3
**Recommendation:** 6

**Strength And Weaknesses:**

Strengths:
1 ) It is interesting to know with a teacher and a single large image, what level of accuracy a student can achieve on certain images.

2 ) It demonstrates (vanilla) knowledge distillation's immense power. It means KD can be done with extremely small amount of data. The paper may consider citing related few-shot KD methods such as [1].

3 ) The paper's ablation study is comprehensive and reveals many interesting points.


Weaknesses

1 ) The teacher is supervised pre-trained. It contains information from a full dataset. Thus it may not be fair to say this study helps to understand the mechanism of NNs when provided with the minimum data. Moreover, the semantic categories are explicitly  modeled with teacher's final classifier, which is clearly learned from large data instead of the single image.

Therefore, I don't think the question in "we go far beyond these and instead ask what the minimal data requirements are for neural networks to learn semantic categories" can be answered with "only one image and augmentations" without mentioning the teacher.

The conclusion sentence in the discussion section may add "together with a supervisory signal provided by a teacher pretrained on large data with many semantic categories, provide sufficient …"

There is another way to frame the conclusion and finding: KD can be done with one image and augmentations to reach decent accuracy. I think this one represents the finding more accurately and concisely.

2 ) As the paper suggested in the intro, the use case of the method is not obvious.

3 ) It may be an exaggeration to say that this is learning from a single datum also because the image used is very high resolution. Small crops of CIFAR-size (32x32) with data augmentations easily surpass thousands. The "image space" doesn't specify image size so in extreme I could tile all ImageNet images into one and still call it one "image".

[1] few sample knowledge distillation for efficient network compression


**Summary Of The Paper:**

The paper proposes a distillation framework that makes a network learn with crops and augmentations from a single image. The learning target comes from a pre-trained teacher's outputs. The results obtained is of decent level (69% ImageNet top1). Various design options and internal properties of the resulting network are analyzed.

**Summary Of The Review:**

The paper presents some interesting findings but I think there are exaggerations in claims as discussed in weaknesses, so both considered my recommendation is "marginally above".

---

> ### Author Response · Authors · 2022-11-07
> **Response to AYLh**
>
> Thank you for your thorough review and your positive feedback!
> Please find the responses to all your points below:
>
> > Therefore, I don't think the question in "we go far beyond these and instead ask what the minimal data requirements are for neural networks to learn semantic categories" can be answered with "only one image and augmentations" without mentioning the teacher.
>
> We wish to stress that we have taken great care to mention the supervisedly trained teacher in many locations: Abstract ("from a supervised teacher"), Figure 1 ("from a pretrained teacher"), Introduction ("the outputs of a supervisedly trained model"), Method ("knowledge of a pretrained teacher") and regarding the Discussion we refer to next point.
> Regarding the explicit categories contained in the classifier, we wish to also refer you to to the transfer learning experiments that we have now conducted in response to reviewer `rSwz`, which shed some light on the feature quality besides the explicit categories.
>
> > The conclusion sentence in the discussion section may add "together with a supervisory signal provided by a teacher pretrained on large data with many semantic categories, provide sufficient …" There is another way to frame the conclusion and finding: KD can be done with one image and augmentations to reach decent accuracy. I think this one represents the finding more accurately and concisely.
>
> This is a great suggestion and will hopefully reduce confusion. Thank you for this, we have incorporated this into our revision. And for convenience, we show the revised final paragraph of the discussion section below (edits *bolded*):
>
> _In this work, we have analyzed whether it is possible to train neural networks to extrapolate to unseen semantic classes **with the help of a supervisory signal provided by a pretrained teacher**. Our quantitative and qualitative results demonstrate that our novel single-image knowledge distillation framework can indeed enable training networks from scratch to achieve high accuracies on several architectures, datasets and domains. **This demonstrates that knowledge distillation can be done with just a single image plus augmentations**, and also raises several further research questions, such as the dependency of the source image and the target semantic classes; how networks combine features for extrapolation; and the role and informational content of augmentations; all of which we hope inspires further research._
>
>
>
>
> > As the paper suggested in the intro, the use case of the method is not obvious.
>
> While we agree that there is presently no direct use case -- our main goal is to conduct an investigation of theoretical and fundamental nature. We believe that shedding light on something as ubiquitous and almost-taken-for-granted as augmentations is of great interest to the ICLR community and might (further down the line) even lead to applications.
>
>
> >  It may be an exaggeration to say that this is learning from a single datum also because the image used is very high resolution.
>
> While definitely higher-res than CIFAR, the `Animals` image, which performs on par with the larger `City` image, is only of size 1300x600px. Importantly, in Table 1, we also compare against the number of pixels and show that there is added benefit from using a single image compared to many more, unrelated images even if they come from the target distribution.
>
> > Small crops of CIFAR-size (32x32) with data augmentations easily surpass thousands. The "image space" doesn't specify image size so in extreme I could tile all ImageNet images into one and still call it one "image".
>
> While a tiled image of multiple images is rare when considering real photographic images -- we agree with the reviewer, that just specifying "a single image" could be specified further. We will clarify this and also wish to note that that is why we further compare against number of pixels (see table 1) and image sizes in MB as JPEG images (Figure 2, Figure 7).
>
>
>
> > The paper may consider citing related few-shot KD methods such as [1].
>
> Thank you for this useful reference. We have added it to our related works section.

---

> > ### Comment · Reviewer_AYLh · 2022-11-17
> > **Thanks for the response**
> >
> > Most of my concerns are addressed to some degree and I think the paper is strengthened. I think the paper is above acceptance threshold.

---

> > > ### Author Response · Authors · 2022-11-17
> > > **Thanks**
> > >
> > > Thanks! We're glad you feel like the paper is strengthened.
> > > You mentioned all concerns are addressed "to some degree" -- if there is anything that need to be further improved, we'd be happy to incorporate any further feedback into the camera-ready. We would be thankful if you could also update your final rating to reflect the improvement in the paper.

---

### Official Review · Reviewer_rxVL · 2022-10-25

**Confidence:** 4
**Correctness:** 3
**Technical Novelty And Significance:** 2
**Empirical Novelty And Significance:** 1
**Recommendation:** 6

**Clarity, Quality, Novelty And Reproducibility:**

The overall presentation and writing sound good to me. The authors also provided the code, so it might be possible to reproduce the results.

**Strength And Weaknesses:**

Strength:
In order to verify its effectiveness, the authors have conducted experiments on different datasets and also conducted ablation studies on different components of this proposed method.


Weakness:

1) The motivation is not clear. Since this proposed method only uses a single image and its augmentations as its training data (it also uses a “teacher” model which should also be considered as an input information source), thus it might be acceptable that it will sacrifice some performance/accuracy. But the question is do we really want to do that? I do not see any benefits from doing this. If we think there are some domains where obtaining training data is expensive, then we say this model may be helpful. However, the thing is we still need a “good” teacher which should be trained on a large amount of dataset in this domain.

2) Using a single image to train a classifier and gain reasonable accuracy sounds surprising; however, there are several recent papers finding that even a “randomly initialized neural network with frozen network” can also yield good classification performance (I list several references below). Given this background, the finding may not be as impressive as it sounds since in any way it has updated its network weights and take the advantage of the knowledge from the “good teacher”.

Ref1: K. Jarrett, K. Kavukcuoglu, M. Ranzato, and Y. LeCun, “What is the best multi-stage architecture for object recognition?” 2009 IEEE 12th International Conference on Computer Vision, Sep. 2009, pp. 2146–2153, ISSN: 2380-7504.

Ref2: H. Zhou, J. Lan, R. Liu, and J. Yosinski, “Deconstructing Lottery Tickets: Zeros, Signs, and the Supermask,” in Advances in Neural Information Processing Systems, vol. 32. Curran Associates, Inc.,
2019.

Ref3: K. Sreenivasan, S. Rajput, J.-y. Sohn, and D. Papailiopoulos, “Finding Everything within Random Binary Networks,” 2021.

Ref4: S. Baek, M. Song, J. Jang, G. Kim, and S.-B. Paik, “Face detection in untrained deep neural networks,” Nature Communications, vol. 12, no. 1, p. 7328, Dec. 2021.

Ref5: G. Kim, J. Jang, S. Baek, M. Song, and S.-B. Paik, “Visual number sense in untrained deep neural networks,” Science Advances, vol. 7, no. 1, p. eabd6127, Jan. 2021.






/********************************** After Rebuttal******************************/

Thanks for your response. After reviewing the responses and the discussions, most of my major concerns have been resolved. I would slightly raise my rating.


**Summary Of The Paper:**

This paper uses a single image along with its augmentation and a pretrained classification model (teacher) together to train a student classifier. Through experiments on different datasets including images, audios, and videos, the authors find that the student classifier can achieve reasonably good classification performance.

**Summary Of The Review:**

Overall, this paper is easy to follow and understand. It also has a lot of experiments. The big issue is it is not clear what benefits this method can bring to the community; also “random neural network with frozen weights” can also achieve impressive classification accuracy, then the finding in this paper does not sound as surprising as it is.

---

> ### Author Response · Authors · 2022-11-07
> **Response to rxVL**
>
> Thank you for your careful review and your time.
>
> > The motivation is not clear. [...] the question is do we really want to do that? I do not see any benefits from doing this.
>
> While this is a valid point when coming from an application-oriented perspective, we believe our motivation is quite clear. As stated in the abstract, introduction and conclusion, our main goal is to conduct an investigation of theoretical and fundamental nature. We believe that despite our findings not having an _immediate_ application, shedding light on something as ubiquitous and almost-taken-for-granted as augmentations, is of great interest to the ICLR community.
>
> > there are several recent papers finding that even a “randomly initialized neural network with frozen network” can also yield good classification performance [...] Given this background, the finding may not be as impressive as it sounds [...].
>
> Firstly, thank you for these very interesting references. We have added them in our revised version. Yet, we wish to make clear the key difference to these works: While Refs1-5 show what surprisingly complex tasks can be accomplished _with partially untrained networks_ *but plenty of training data*, we show the flip-side of this, namely that we can _train networks fully_ *with very little data* as input. Further differences to these works also include that we experiment on far more complex datasets than only MNIST or CIFAR,  and even go beyond the image-modality to audio and video.
>
> We hope that this has cleared any remaining doubts and look forward to discussing further.

---

> > ### Comment · Reviewer_rxVL · 2022-12-05
> > **Thanks for your response**
> >
> > Thanks for your response. After reviewing the responses and the discussions, most of my major concerns have been resolved. I would slightly raise my rating.

---

### Official Review · Reviewer_rSwz · 2022-10-25

**Confidence:** 4
**Correctness:** 3
**Technical Novelty And Significance:** 3
**Empirical Novelty And Significance:** 3
**Recommendation:** 8

**Clarity, Quality, Novelty And Reproducibility:**

The paper is well-written and easy to follow. It provides a comprehensive investigation of the phenomenon and covers most of the important aspects. The experiments are well-documented and the details should suffice to reproduce the results. To my knowledge, distillation based on a few or a single image has not been explored before, so the contributions are novel as far as I can tell.

**Strength And Weaknesses:**

Overall I feel that the paper provides a comprehensive investigation of single/few image based distillation. Nevertheless, I have a few suggestions on how to improve the paper.

*Strengths:*
- The observation that a single or a few images are enough for distillation is surprising and interesting (at least to me).
- The paper covers a lot of datasets, architectures, and modalities, and provides many insightful ablations, and as such answers many important questions on the phenomenon.

*Weaknesses/suggestions for improvement:*
- One important open question: What happens to transfer and robustness properties of the distilled models? The output confidence scores seem to change significantly on the testing set (Figure 3), and the paper also delivers a potential hint why this could be happening: The output probabilities on the samples used for training are very flat for the teacher (Figure 8). Given this observation, the models might be mis-calibrated or have reduced OOD robustness, which would merit more investigation.
- A related analysis would be to look at examples for which are misclassified by the student but correctly classified by the teacher. Are there some patterns.
- It would be interesting to see if more sophisticated synthetic data than uniform noise can lead to better transfer, such as the synthetic data sets considered in Baradad et al. 2021.
- The observation that the method works less well for transformer models could be discussed in the main paper. Given that transformers are harder to train on small data sets than ConvNets this is not very surprising.


**Summary Of The Paper:**

The paper investigates knowledge distillation using a data set consisting of a single or a few images which are heavily augmented. The paper finds that this type of knowledge distillation is as effective as using the training set, at least for small scale data sets such as CIFAR-10/100. Different design choices such as student and teacher architecture, the effect of the teacher training set as well as the image(s) chosen for distillation and the augmentations used are explored in-depth. Furthermore, the method is also shown to be effective for other modalities such as audio and videos, albeit these are less explored than images.

**Summary Of The Review:**

The paper provides an interesting and comprehensive investigation of knowledge distillation based on a single or a few images. I think the paper should be published, but I also believe the points I raised above regarding calibration/robustness should be answered before publication, which is why I chose my rating defensively.

---

> ### Author Response · Authors · 2022-11-07
> **Response to rSwz (part 1)**
>
> Thank you for the thorough review and the useful feedback! Please find our answers to the points raised below.
>
> > What happens to transfer and robustness properties of the distilled models?  [...] the models might be mis-calibrated or have reduced OOD robustness, which would merit more investigation.
>
> Indeed, this is a very interesting question. We have now investigated both transfer performance and and OOD robustness.
>
> ### Transfer
> In order to compare transfer performance, we experiment on typical representation learning downstream tasks: linear probing on ImageNet and Places, as well as low-shot classification results on Places and Pascal VOC.
>
> In order to disentangle the effect of our method and the training data, we further trained a self-supervised method on the single-image generated dataset. For this we have simply run MoCo-v2 out-of-the-box on our dataset for 200 epochs, which we will refer to as "1-image MoCo-v2".
>
> **Linear probing**
>
> The first transfer learning we conduct is linear probing, where a linear layer is trained on top of the frozen backbone.  Numbers in [brackets] indicate no training (for the IN1k -> IN1k case).
>
> | | IN1k | Places205|
> |  --  | -| -|
> | IN-1k R50 |[76.2] | 51.5 |
> | IN-1k R18 (teacher) |[69.8]| 44.1 |
> | 1-image MoCo-v2 | 28.5 | 28.8 |
> | 1-image Distill | 68.8 | 47.2 |
>
> **Low-shot classification**
>
> Next, we conduct low-shot classification by training a SVM on top of the frozen representation with limited numbers of labelled images (4 or 16 per class (p.c.)).
>
> | | PVOC (4 p.c.) |PVOC (16 p.c.)| Places205 (4 p.c.)| Places205 (16 p.c.) |
> |  --  | -| -| -| -|
> | IN-1k R50 | 73.8 | 82.3 | 27.0 | 35.4 |
> | IN-1k R18 (teacher) | 65.7 | 77.1 | 21.4 | 30.4 |
> | 1-image MoCo-v2 | 17.2 | 26.6 | 4.5  | 9.7  |
> | 1-image Distill | 58.9 | 73.5 | 21.0 | 31.3 |
>
> _Result from transfer experiments:_
> From both of these tables we find that the single-image trained model matches that of the teacher for the tasks considered, while the 1-image MoCo model lacks behind. This shows that the training data alone is insufficient without the strong signal from the teacher for transfer performances.
>
>
>
>
> ### Robustness benchmarks
>
> For robustness, we have additionally evaluated our models on CIFAR-100-C and ImageNet-C [1]. We show results for the latter here:
>
> |                         |       |         |        Noise        |       Noise            |   Noise                   |         Blur         |         Blur           |    Blur                 |      Blur             |      Weather      |    Weather                |    Weather              |             Weather        |        Digital        |          Digital            |     Digital               |         Digital          |
> |:-----------------------:|:-----:|:-------:|:-------------------:|:-----------------:|:----:|:--------------------:|:------------------:|:-----:|:-----------------:|:-------------:|:--------:|:------:|:-------------------:|:---------------------:|:--------------------:|:------------------:|:-----------------:|
> | Network                 | Error | mCE | Gauss. | Shot | Impulse | Defocus | Glass | Motion | Zoom | Snow | Frost | Fog | Bright | Contrast | Elastic | Pixel | JPEG |
> | IN-1k R18 (teacher)     |  30.2 |   84.7  |          87         |         88        |          91          |          84          |         91         |          87         |         89        |         86        |         84         |        78        |          69         |           78          |          90          |         80         |         85        |
> | 1-image R50x2 (student) |  31.0 |   85.9  |          88         |         89        |          91          |          85          |         92         |          88         |         89        |         88        |         86         |        82        |          71         |           80          |          91          |         82         |         87        |
> | 1-image R50 (student)   |  33.8 |   89.8  |          93         |         93        |          96          |          87          |         94         |          90         |         92        |         91        |         90         |        84        |          77         |           83          |          96          |         87         |         93        |
>
> From the above table (which reports normalised error rates) we find that the overall corrupted error (mCE) level only slightly increases by 1.2% when comparing our best student model with the IN-1k trained teacher model. However, for the slightly worse R50 student model (which only has 66.2% Top-1 accuracy on clean ImageNet) the mCE measure increases by 5.2% showing that it suffers more from image corruptions than the teacher.  It is also evident even in the R50x2 model that its robustness is worse on every corruption type compared to the teacher, showing how robustness is clearly limited by the teacher's performance. CIFAR-100-C findings mirror these results.

---

> > ### Author Response · Authors · 2022-11-07
> > **Response to rSwz (part 2)**
> >
> > > A related analysis would be to look at examples for which are misclassified by the student but correctly classified by the teacher. Are there some patterns.
> >
> > Thank you for this interesting idea
> > For our ImageNet model, we have investigated which classes the student drastically underperforms and overperforms compared to the teacher model:
> >
> > **overperforming top 10 classes:**
> > `tobacco shop, trimaran, notebook, stingray, gong, gazelle, alp, jeans, analog clock, Siberian husky`
> > (ranging from +10% to +22% compared to teacher)
> >
> > **underperforming top 10 classes:**
> > `Tibetan terrier, Lakeland terrier, golden retriever, dhole, Border terrier, cocker spaniel, Welsh springer spaniel, Brabancon griffon, collie, mountain bike` (ranging from -34% to -20% compared to teacher)
> >
> > *Result:* What is evident, especially from the underperforming classes is that 9/10 of these are dog breeds. This shows how our single-image trained model lacks behind in very fine-grained classification. Likely this is because the single image lacks the necessary structures and patterns for disambiguating these classes despite the help of augmentations.
> >
> >
> > Apart from this qualitative analysis, we have further added a more quantitative evaluation.
> > In particular, we have analysed if there is a relationship between the student's per-class performance with the teachers training signals' strength per-class.
> > For this, we plot the student validation performance against the frequency of how often a particular class appears as a top-1 prediction of the teacher during 1 epoch of training (this roughly indicates how frequent a 'likeness' of a class is seen in the training data).
> > As shown in Figure 18 of the revised paper, we find that it is unrelated (R-squared=0.07), ie even classes that rarely appear as top-1 prediction from the teacher achieve high student performances. This also shows the student is profiting heavily from the knowledge contained in the soft-predictions. We further compare these per-class performances against the teacher's performance in Fig. 19 and find again no apparent relationship (R-squared=0.01) between when the student under or overperforms the teacher vs how often a particular likeness of a class appears.
> >
> >
> >
> >
> > > It would be interesting to see if more sophisticated synthetic data than uniform noise can lead to better transfer, such as the synthetic data sets considered in Baradad et al. 2021.
> >
> > We already have results from Baradad et al in our Table 2 using their best generation method ("StyleGAN").
> > As this synthetic dataset was already underperforming by around 6% for CIFAR-100, we had not run it for ImageNet, but have done so now:
> >
> > |          Input     | Teacher |    | Student | 10   | 20   | 30   | 50   | 200  |
> > |---------------------|-----|--------|-----|---|---|---|---|---|
> > | Noise Image    | R18     | → | R50     | 0.1  | 0.1  | 0.1  | -    | -    |
> > | Random Uniform [0,1] Noise    | R18     | → | R50     | 0.1  | 0.1  | 0.1  | -  | -    |
> > | StyleGAN dataset from Baradad et al.  | R18     | → | R50     | 15.4  |   30.1 | 37.8  | 44.4  | -    |
> > | Bridge Image      | R18     | → | R50     | 21.2 | 34.8 | 40.0 | - | - |
> > | City Image      | R18     | → | R50     | 34.5 | 47.0 | 52.2 | 56.8 | 66.2 |
> >
> > In the Table above we compare the large-scale dataset of 1.3M images from Baradad et al. to noise baselines and two single image ones from our paper.
> > We find that the results agree with those conducted on CIFAR-100: The synthetic dataset of Baradad et al. outperforms random noise or a noise-image, yet still falls behind a single, real image (even a less dense one, such as the Bridge image). Note also that the dataset of Baradad is 1.3M images.
> >
> >
> > > The observation that the method works less well for transformer models could be discussed in the main paper. Given that transformers are harder to train on small data sets than ConvNets this is not very surprising.
> >
> > Thanks, we will add this point to the main paper.
> >
> > >  I think the paper should be published, but I also believe the points I raised above regarding calibration/robustness should be answered before publication, which is why I chose my rating defensively.
> >
> > We understand. We hope that our detailed response with various further evaluations has now cleared up any doubts and hope that this better version of our paper will be reflected in your recommendation.
> >
> > ### References
> > [1] Benchmarking Neural Network Robustness to Common Corruptions and Perturbations. Hendrycks and Dietterich. ICLR 2019

---

> > > ### Comment · Reviewer_rSwz · 2022-11-17
> > > **Response to author response**
> > >
> > > I thank the authors for the additional results, which I believe add valuable new insights to the paper. I therefore raised my score accordingly.

---

### Author Response · Authors · 2022-11-07
**Revised version**

We thank all the reviewers for their comments and useful feedback. We have just uploaded a revised version of the paper that includes the points that we have also mentioned in the response to each reviewer.
For convenience, we report the main differences to the previous version here:

* Added transfer learning experiments of linear probing on ImageNet, Places and low-shot classification on Pascal and Places, see Sec. B11
* Added robustness experiments on ImageNet-C, see Sec. B12
* Added per-class analysis of student compared to teacher (qualitative and quantitative), see Sec. B13
* Added further ImageNet-level distillation results comparing with Baradad et al., see Sec. B4
* Added related works as suggested by `rxVL` and `AYLh`
* Added reference to ViTs in main paper as suggested by `rSwz`
* Adapted claims & language, especially in the discussion section as suggested by `AYLh`

---

### Author Response · Authors · 2022-11-15
**Final three days**

Dear Reviewers, dear AC,

We're entering the last three days of the discussion period.
8 days ago we have responded to each reviewer's points and uploaded a revised version of our paper which contains multiple new experiments and evaluations.

We would be thankful if the reviewers could take a look at our responses and update their review if we have cleared up the remaining concerns. If not, we're happy to discuss further!

---

### Author Response · Authors · 2022-11-18
**End of discussion period comment from the authors**

We would like to thank the reviewers again for their time and effort in reviewing and the positive feedback we have received.

In our final revised version we have further improved the presentation of figures in the appendix and adjusted spacing so the main paper is back to 9 pages (as the last version was one line above).

To conclude, we are happy that our clarifications, additional experiments and evaluations in the revised version (see [post below](https://openreview.net/forum?id=6kxApT2r2i&noteId=hN9DaOgNW7) for the overview) have yielded *valuable new insights* (`rSwz`, who raised their score from 6->8) and that *concerns are addressed*  (`AYLh`) and we agree that this version now contains even more insights at hopefully a higher clarity of presentation. Thanks again.

---

### Public Comment · ~Wenhai_Wan1 · 2023-03-28
**It seems that there is some text overlap on the eighth page of the paper, which appears to be quite distracting for reading.**

It seems that there is some text overlap on the eighth page of the paper, which appears to be quite distracting for reading.

---

> ### Author Response · Authors · 2023-04-04
> **Thanks**
>
> Thanks for noticing and sorry for the inconvenience.
> Here's a clear [PDF version](https://arxiv.org/pdf/2112.00725.pdf) from [arxiv](https://arxiv.org/abs/2112.00725).

---

### Decision · Program_Chairs · 2023-01-20

**Decision:**

Accept: poster

**Justification For Why Not Higher Score:**

Unclear interpretation of results.

**Justification For Why Not Lower Score:**

Interesting result. Clear presentation.

**Metareview: Summary, Strengths And Weaknesses:**

In this work, the authors demonstrate the ability to semantically classify images based on a single example by leveraging heavy data augmentations. In particular, the authors employ a single image along with its augmentations and a pretrained classification model to train a student classifier with knowledge distillation. The authors find that a single or a few images are sufficient to achieve good performance on CIFAR-10 and CIFAR-100. The authors show additional results on audio and video datasets.

The reviewers applauded the author’s comprehensive experiments and the clarity of presentation. In addition, some reviewers commented that they were surprised about the quality of results given the limited amount of data. In follow up discussion, the authors offered some new results suggesting that the few image distillation were sufficient to even produce non-trivial results on transfer learning and robustness experiments. The reviewers did find some concern about the interpretation of the results and asked the reviewers to clarify their writing accordingly. These issues, while important, have been addressed and provide for a reasonable interpretation of the contributions of this work and the placement of this work within the literature. Given the interesting findings based on convincing experiments that are clearly explained, this paper will be accepted in this conference.


**Note From Pc:**

if the above contains the word "oral" or "spotlight" please see: "oral" presentation means -> notable-top-5% and "spotlight" means -> notable-top-25%. As stated in our emails, we are disassociating presentation type from AC recommendations